# Embodied Referring Expression Comprehension Through Multimodal Residual Learning

## Abstract

Comprehending embodied interactions within real-world settings poses a considerable challenge, attributed to the multifaceted nature of human interactions and the variability of environments, necessitating the development of comprehensive benchmark datasets and multimodal learning models. Existing datasets do not adequately represent the full spectrum of human interactions, are limited by perspective bias, rely on single viewpoints, have insufficient nonverbal gesture capture, and have a predominant focus on indoor settings. To address these gaps, we present an Embodied Referring Expressions dataset (called Refer360), which contains an extensive collection of embodied verbal and nonverbal interaction data captured from various viewpoints across various indoor and outdoor settings. In conjunction with this benchmark dataset, we propose a novel multimodal guided residual module (MuRes) that helps the existing multimodal models to improve their representations. This guided residual module acts as an information bottleneck to extract salient modality-specific representations, and reinforcing these to the pre-trained representations produces robust complementary representations for downstream tasks. Our extensive experimental analysis of our benchmark Refer360 dataset reveals that existing multimodal models alone fail to capture human interactions in real-world scenarios comprehensively for embodied referring expression comprehension tasks. Building on these findings, a thorough analysis of four benchmark datasets demonstrates superior performance by augmenting MuRes into current multimodal models, highlighting its capability to improve the understanding and interaction with human-centric environments. This paper offers a benchmark for the research community and marks a stride towards developing robust systems adept at navigating the complexities of real-world human interactions.

## 1 Introduction

An understanding of embodied interaction by combining verbal messages and nonverbal signals is crucial for robots in achieving fluent collaboration with people in human environments McNeill (2012); Arbib et al. (2008); Liszkowski et al. (2006; 2004); Tomasello (2010); Tang et al. (2020); Stacy et al. (2020); Kratzer et al. (2020); Islam and Iqbal (2020; 2021). It enables their smooth integration into human teams and facilitates more natural interactions with people Chen et al. (2021); Islam et al. (2024a; 2022a); Kratzer et al. (2020); Yasar* et al. (2022); Yasar and Iqbal (2021). However, comprehending multimodal cues by extracting and fusing representations from verbal and non-verbal signals poses some significant challenges Samyoun* et al. (2022); Islam et al. (2022b); Feichtenhofer et al. (2019). Moreover, these difficulties are exacerbated by inherent data collection biases, which result in a nuanced yet restricted comprehension of human behaviors and interactions due to environmental constraints, pre-defined human-robot interactions, and the diversity of sensory modalities Islam et al. (2024a). These limitations underscore the need for a robust multimodal model to extract complementary representations trained on a diverse dataset.

Existing datasets, such as YouRefIt Chen et al. (2021) and MoGaze Kratzer et al. (2020), while capturing real-world embodied interactions, have crucial limitations that challenge the development of robust comprehension models. First, these datasets contain verbal utterances from the speaker's or observer's perspective, such as "left ball" versus "right ball". This bias in the trained data limits the models' ability to understand embodied interactions comprehensively. Second, the reliance on

Figure 1: Refer360 data collection setup to capture human interactions using Azure Kinect mounted on the robot and a Pupil Smart Glass worn by the subject (left). Interaction frames from three different views (Exo, Ego, and Exo). Highlighting the canonical frames, i.e., frames where the subject precisely points to an object (right).

single-view (exo or ego) data collection introduces view bias, limiting model performance across diverse environments. Multi-view data capturing (ego, exo, and top views) is essential for overcoming occlusions in object visibility and interaction nuances, thereby enabling a more holistic understanding of embodied interactions. Third, existing datasets partially capture nonverbal gestures. These datasets capture either pointing gestures or gazes. However, in embodied interactions, both signals provide complementary information to comprehend an interaction robustly. Fourth, existing datasets are collected indoors and in constrained settings where humans are specifically instructed. Additionally, these datasets are collected from a stationary camera from a fixed angle. These drawbacks in the datasets limit the trained models to comprehend real-world human interactions in diverse and unconstrained settings. A comparison of the existing datasets is given in Table 1.

To address these issues, we have curated a comprehensive and diverse dataset, called Refer360, to facilitate the understanding of human interactions in real-world settings. We have collected the dataset across various indoor and outdoor settings with varying attributes, such as variable lighting conditions, object arrangements, and environment appearances. Our data collection system is depicted in Fig. 1. We have collected multimodal data using a range of sensors to capture interactions comprehensively, including ego and exo visual views, depth, skeleton, infrared, audio, gaze, and pupil tracking. Finally, this dataset contains scenes and verbal utterances annotated by expert human annotators. Data collection was conducted under an approved Institutional Review Board (IRB) protocol.

Beyond dataset biases, another significant challenge in comprehensively understanding embodied referring expressions is the extraction of complementary representations from multimodal data. While existing multimodal models fuse multimodal representations from the frozen pre-trained encoders, leading to performance enhancements across various tasks, the representation gap between these frozen representations can lead to sub-optimal multimodal representations. Several approaches have been proposed in the literature to reduce the representation gap Alayrac et al. (2022); Li et al. (2022; 2023); Liu et al. (2023). However, fusing these frozen representations using self-attention or cross-attention approach can overlook modality-specific cues, limiting the model's ability to effectively leverage and integrate the distinct, complementary cues in multimodal interaction signals (verbal and non-verbal). Thus, extracting salient representations across modalities can help to extract complementary representations.

To address this challenge, we introduce a novel multimodal guided residual module, MuRes, to learn complementary multimodal representation. Unlike existing approaches, MuRes not only extracts aligned representations but also learns modality-specific cues through guided residual connections. Following the information bottleneck principle Islam et al. (2023); Wang et al. (2022); Tishby and Zaslavsky (2015); Shwartz-Ziv and Tishby (2017); Tishby et al. (2000); Sun et al. (2022); Alemi et al. (2016); Träuble et al. (2022); Islam et al. (2024b), we design MuRes as a representation bottleneck to extract relevant representations across modalities. Reinforcing these relevant representations can help to extract complementary multimodal representations. This method ensures that the model captures aligned and modality-specific representations across modalities. This complementary fused representation can help comprehensively understand multimodal embodied interactions. Our pro-

Table 1: Comparison of the QA datasets. Existing VQA and EQA datasets do not contain nonverbal gestures (NV), multiple verbal (V) perspectives (MP), contrastive (C), and ambiguous (A) data samples, and outdoor scene data. ‡Embodied (E) interactions refer to humans interacting using multimodal expressions. †Embodied interactions refer to an agent navigating in an environment. *Sythetic Environment. **Please check the supplementary for a detailed comparison with other related datasets.**

| Datasets | V | NV | E | MP | Views | | C | A | Image Frames | Interaction Samples | Environment | Type |
| --- | --- | --- | --- | --- | --- | --- | --- | --- | --- | --- | --- | --- |
| | | | | | Exo | Ego | | | | | | |
| VQA Antol et al. (2015) | ✓ | ✗ | ✗ | ✗ | ✓ | ✗ | ✗ | ✗ | 204K | 614K | Internet | Image |
| GRiD-3D* Lee et al. (2022) | ✓ | ✗ | ✗ | ✗ | ✓ | ✗ | ✗ | ✗ | 8K | 445K | Simulated | Image |
| EQA† Das et al. (2018) | ✓ | ✗ | ✓† | ✗ | ✗ | ✓† | ✗ | ✗ | 5K | 5K | Simulated | Interactive |
| MT-EQA† Yu et al. (2019) | ✓ | ✗ | ✓† | ✗ | ✗ | ✓† | ✗ | ✗ | 19K | 19K | Simulated | Interactive |
| CAESAR-XL‡* Islam et al. (2022a) | ✓ | ✓ | ✓ | ✓ | ✓ | ✓ | ✓ | ✓ | 841K | 1M | Simulated | Image |
| EQA-MX‡* Islam et al. (2024a) | ✓ | ✓ | ✓ | ✓ | ✓ | ✓ | ✓ | ✓ | 750K | 8K | Simulated | Image |
| YouRefIt Chen et al. (2021) | ✓ | ✓ | ✓ | ✗ | ✗ | ✓ | ✗ | ✗ | 497K | 4K | Indoor | Video |
| Refer360‡ | ✓ | ✓ | ✓ | ✓ | ✓ | ✓ | ✗ | ✓ | 1.3M | 14K | Indoor+Outdoor | Video |

posed guided residual module can be used as an adapter module in existing multimodal models to extract salient representations.

To evaluate the effectiveness of our module, we conduct extensive experimental analysis on our Refer360 dataset for comprehending referring expressions, alongside various visual question-answering (VQA) datasets. Furthermore, we have integrated MuRes into existing multimodal models to show the effectiveness of utilizing MuRes for extracting salient complementary multimodal representation. Our experimental analysis suggests that MuRes helps to improve these multimodal models' performance for various question-answering tasks. For example, integrating MuRes improved the CLIP model's performance (IOU-25) by $3.4\%$ and $4.99\%$ on the Refer360 and CAESAR-PRO datasets, respectively. Additionally, MuRes boosted the VQA task's accuracy of VisualBERT model on the ScienceQA Lu et al. (2022) dataset by $4.58\%$ and ViLT Kim et al. (2021) model on the A-OKVQA dataset by $2.86\%$. These performance improvements depict the significance of our proposed guided residual model for extracting complementary multimodal representations for various downstream tasks.

## 2 RELATED WORK

**Embodied Referring Expression Datasets:** In the literature, embodied interactions are studied in two forms. The first involves agents navigating an environment to gather visual data following verbal instructions Das et al. (2018); Yu et al. (2019). The second focuses on comprehending referring expressions involving verbal and nonverbal cues, where agents interpret and respond Chen et al. (2021); Islam et al. (2022a;c). We explore the second aspect of embodied interactions, focusing on understanding multimodal referring expressions.

Several datasets have been curated in the literature to study embodied referring expressions (E-RFE). For example, Chen Chen et al. (2021) developed an embodied referring expressions dataset where a human refers to an object using verbal and pointing gestures. In their proposed dataset, Kratzer Kratzer et al. (2020) mainly focused on capturing the human body motion and eye gaze. To incorporate both verbal and nonverbal signals, Islam Islam et al. (2022a) developed a synthetic dataset by generating nonverbal cues (pointing gesture and gaze) in a virtual environment and template-based verbal instructions. While these datasets demonstrated the importance of developing diverse datasets towards comprehensively understanding of E-RFE, they predominantly focus on indoor settings Chen et al. (2021), static camera view without motion Chen et al. (2021); Kratzer et al. (2020); Islam et al. (2022a;c; 2024a), scripted human interactions Islam et al. (2022a;c; 2024a), limited sensor modalities Chen et al. (2021); Kratzer et al. (2020), and synthetic environments Islam et al. (2022a;c; 2024a). Therefore, these datasets provide limited data samples for developing models for a comprehensive understanding of E-RFE.

**Multimodal Representation Learning:** There has been significant progress in the last several years on developing multimodal models, particularly focusing on Visual Question Answering (VQA) tasks Li et al. (2019); Lu et al. (2019); Kim et al. (2021); Radford et al. (2021); Li et al. (2022; 2023); Zhai et al. (2022); Alayrac et al. (2022); Liu et al. (2023); Goyal et al. (2017); Gao et al. (2015); Yu et al. (2015); Zhu et al. (2016); Krishna et al. (2017). For example, VisualBERT Li et al. (2019) used a Transformer with Self-Attention to extract salient multimodal representation, which

is trained using visually grounded language model objectives. ViLT Kim et al. (2021) processed visual inputs holistically, learning visual-language representations without relying on the regional supervision typically associated with object detection. BLIP-2 Li et al. (2023) designed Querying Transformer to bootstrap vision-language representation from a frozen image encoder. These models achieved performance improvement on VQA tasks by utilizing representation alignment-based training objectives. However, as these objectives primarily focus on representation alignment, the model can not effectively fuse the modality-specific representations. Additionally, utilizing the self and cross-attention approaches primarily focuses on alignment to calculate attention score; hence, complementary representations can not be extracted, which are crucial for comprehensively understanding the multimodal referring expressions.

# 3 DATA COLLECTION

## 3.1 DATA COLLECTION SYSTEM

The goal of the Refer360 dataset is to study real-world human-robot interactions in which a human provides object-referencing instructions to robots across diverse environments, spanning controlled laboratory setups to outdoor locations. To achieve this, we have developed a data collection system that synchronously captures multimodal data of embodied interactions in lab and outside-lab environments, utilizing an Azure Kinect DK azu and a Pupil Glass eye tracker pup. It is worth noting that by 'outside-lab environment,' we encompass settings, including home, outdoor locations, etc.

Figure 1 depicts a sample data collection setup of Refer360. The Azure Kinect DK is mounted on an Ohmni telepresence robot ohm to incorporate camera motion and replicate real-world settings. The Kinect sensor offers multiple data streams that capture different interaction modalities. Its RGB camera continuously records visual data, providing an external or ex-

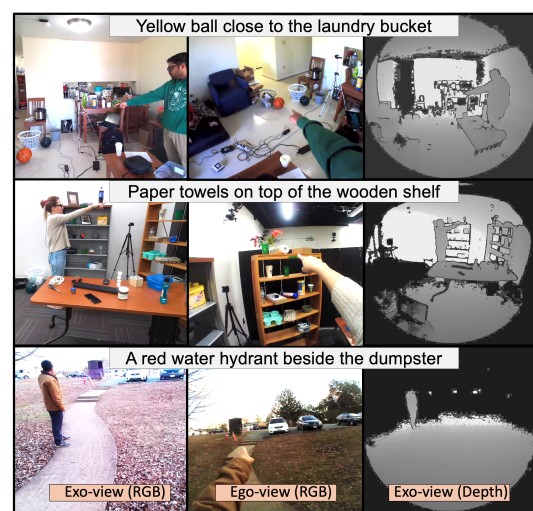

Figure 2: Sample canonical frames from Refer360 dataset in three different views: Exo-view (RGB), Ego-view (RGB), and Exo-View (Depth). The first, second, and third rows contain interaction samples from a home, lab, and outdoor location.

ocentric perspective of the participant's actions. The Pupil eye tracker records an RGB data stream, capturing the participant's first-person or egocentric perspective. Additionally, the Kinect sensor captures depth, infrared, and audio data streams, enabling analysis of the participant's environment and audio cues. We utilize Kinect's Body Tracking SDK Microsoft to capture 3D skeletal data with 32 body joints, allowing us to track the participant's movements and postures. By combining exocentric and egocentric viewpoints, along with multimodal data from the same interaction, our system offers a comprehensive understanding of embodied human-robot interactions.

We have developed a Python-based application to synchronize the data collection process. It utilizes the pyKinectAzure Gorordo (Year of access) library for the Kinect sensor's data streams and Pupil Labs' Real-time Python API Pupil Labs (Year of access) for the Eye Tracker's data streams. We log the UNIX timestamps of data capture events for multiple sensor data streams from Kinect and Eye Tracker. We used these timestamps to synchronize the captured data during post-processing. This timestamp-based synchronization method can be extended to seamlessly integrate various additional sensors for enhanced functionality and versatility. We will opensource this data collection system for future research. **Details of the data collection system can be found in Appendix A.**

## 3.2 PARTICIPANTS

After receiving approval from the Institutional Review Board (IRB) for our study involving human participants, we recruited 66 participants for the study and data collection with 53% males ($n = 35$) and 47% females ($n = 31$). The participants were primarily students from various academic backgrounds. The average age of the participants was 26.66 years, with a standard deviation of 3.36 years. One participant did not consent to release the data. We excluded that participant data from

Table 2: Statistical breakdown of Refer360 dataset.

|  | Sessions | Interactions | Frames | Canonical Frames | Avg. Interaction Duration | Total Duration |
|---|---|---|---|---|---|---|
| Lab | 198 | 10,814 | 2,472,939 | 22,356 | 4.484 sec | 13.48 hr |
| Outside-lab | 194 | 3,176 | 759,018 | 6,380 | 4.691 sec | 4.14 hr |
| **Total** | **392** | **13,990** | **3.2M** | **28,736** | **4.531 sec** | **17.62 hr** |

Refer360. Each participant was compensated $15 for 1 hour of their time, which is higher than the state minimum wage guideline.

### 3.3 DATA COLLECTION PROTOCOL

All data collection tasks required participants to provide object referencing instructions across different sessions, where the environment setup, objects, and data capturing viewpoints varied. Before beginning the study, participants reviewed consent documents and task instructions. They then completed a pre-task survey, providing demographic information and details about their experience with robots. Next, participants wore the eye tracker and participated in the data collection sessions. These sessions occurred under one of two distinct conditions: constrained or unconstrained. In the constrained condition, participants received guidelines on the instruction format and were encouraged to utilize verbal and non-verbal modalities for natural interaction. Conversely, subjects received no specific instruction format or modality suggestions in the unconstrained condition. After completing all sessions, participants completed a post-task survey indicating their preferred method of object referencing. The options provided were using only verbal instructions, only gestures, or a combination of verbal instructions and gestures. Participants also signed a consent form permitting the release of the collected dataset. Please refer to Appendix A for further details on the data collection protocol and procedure. The study protocol was approved by the University of Virginia's IRB.

### 3.4 DATASET POST-PROCESSING

We have recorded a single video file utilizing the Kinect sensor for each session, which contains three data streams: RGB, Depth, and Infrared. Using the data collection application, we read the Kinect sensor's IMU and 3D skeleton joint data and stored them in separate JSON files. We utilize the FFmpeg ffm library to split the Kinect video stream into three separate streams for RGB, Depth, and Infrared. The IMU time series data is split into two files: accelerometer readings and gyroscope readings. We extracted the recorded audio from Kinect as an MP3 file. For each session, the Pupil eye tracker generates a video file in MP4 format and saves it to the Pupil Cloud with event timestamps.

One of the major challenges in the data post-processing was to synchronize the Azure Kinect and Pupil Eye Tracker data and segment each interaction. We used each interaction's start and end times for the segmentation from the Pupil Cloud event timestamps log. Additionally, we logged canonical frames (Figure 1 (right)), i.e., frames where participants precisely pointed to the object of interest during data collection. We leveraged the FFmpeg library to split the data into individual interactions and these specific canonical frames for Kinect and eye-tracking data. We used the Pupil Labs' Real-time Python API for the eye tracker to access the corresponding recordings stored in the Pupil cloud, matching them to the Kinect data using timestamps. Finally, we employed the OpenAI Whisper OpenAI (Year of access) library to transcribe the audio data captured by the Kinect. Under the approved IRB, five human experts validated all interaction segmentation, synchronization, and audio transcriptions to ensure high-quality data. This dataset was annotated by human annotators from an external company, which provides data annotation services. Figure 2 illustrates sample interactions from Refer360 dataset along with the audio transcription.

## 4 DATASET ANALYSIS

Table 2 presents a detailed statistical breakdown of our Refer360 dataset. The data collection phase involved 392 sessions split between lab and outside-lab environments. A total of $13,990$ interactions were recorded within $17.62$ hours of recording time. A total of $14,368$ frames were captured. There were approximately $36.65$ frames in each session. The average session length was $2.69$ minutes, and each interaction lasted $4.53$ seconds on average.

To gain insight into participants' preferred methods of object referencing, we analyzed the post-task survey data. The results revealed that an overwhelming majority of participants, 96.97% (n = 63), preferred using a combination of verbal instructions and non-verbal gestures, such as gaze and pointing. Only a small fraction, 3.03% (n = 2), preferred using verbal instructions alone. Interestingly, none of the participants chose to rely solely on non-verbal gestures as their preferred method of communication. These findings highlight the strong preference for combining verbal and non-verbal cues when referencing expressions in embodied settings.

## 5 MURES: MULTIMODAL GUIDED RESIDUAL MODULE

The task of grounding objects, referred to by embodied interactions, requires a comprehensive understanding of verbal utterances and nonverbal gestures. Existing visual-language (VL) models often utilize pre-trained frozen encoders to extract visual and language representations, fusing using self-attention or cross-attention approaches for downstream task learning. These fusion approaches can lose salient information due to the modality gap between frozen language and visual representations, resulting in sub-optimal multimodal representations and decreased downstream task performance. To prevent this from happening, one of the prevalent approaches is to utilize a residual connection, which can improve gradient flow Huang et al. (2016; 2017); He et al. (2016) and reinforce a prior representation. However, residual connections contain no information bottleneck, resulting in visual and language representations that contain unrelated information for downstream tasks. From this motivation, we design a multimodal guided residual module, MuRes, to reinforce salient multimodal representations for downstream tasks (Fig. 3).

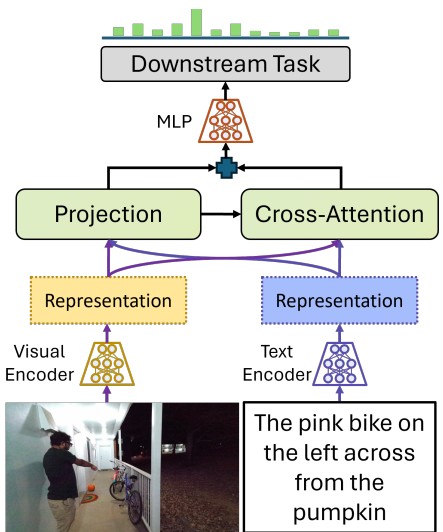

Figure 3: Multimodal Model, MuRes, with the Guided Residual module. Visual and language representations are extracted and projected from a pre-trained VL model. The projected representations are fed into the cross-attention module as the query. The key and value are the original extracted visual and language representations on the residual connection. The output from the cross-attention module and projection are summed for downstream task learning.

**Visual-Language Representations:** Similar to existing models Alayrac et al. (2022); Li et al. (2022; 2023); Zhai et al. (2022); Kim et al. (2021), we first extract visual and language representations using a frozen pre-trained encoder. We used state-of-the-art VL models to extract visual ($V \in \mathbb{R}^{D_V}$) and language $L \in \mathbb{R}^{D_L}$ representations, such as CLIP Radford et al. (2021), DualEncoder Wu et al. (2019), ViLT Kim et al. (2021), and BLIP-2 Li et al. (2023). Here, $D_V$ and $D_L$ are the dimensions of visual and language representations from the pre-trained encoders.

**Multimodal Guided Residual Module:** We introduce a multimodal guided residual module to reinforce salient portions of modality-specific representations, serving as an information bottleneck over vanilla residual connection He et al. (2016) reinforcing entire representations. This is done by focusing on the most relevant parts of the visual or language representations using cross-attention. Cross-attention is similar to self-attention but has a crucial difference in its inputs. In cross-attention, the query is different from the keys and values, whereas in self-attention these are the same. This allows for the usage of projected visual ($V^p$) and language ($L^p$) representations as the query ($q$), and usage of the originally extracted visual ($V$) and language ($L$) representations as the key ($k$) and value ($v$):

$$\{V^g, L^g\} = \text{Cross-Attention}(q = \{V^p, L^p\}, k = \{V, L\}, v = \{V, L\}) \tag{1}$$

This design allows for maintaining beneficial aspects of residual connections, such as improved gradient flow and reinforcement of prior representations, while establishing an information bottleneck on the residual connection. After extracting the guided residual representations, they are added to the projected representations as in vanilla residual connections: $V^f, L^f = V^p + V^g, L^p + L^g$. Finally, we fused these representations ($V^f, L^f$) for downstream task learning.

**Training Model:** To demonstrate the MuRes model's effectiveness at improving representations, we train for two downstream tasks: comprehending embodied referring expressions designed as an object bounding box prediction and visual-question answering designed as a multiple choice question-answering task. We used a regression loss for the object bounding box prediction task and a classification loss for the multiple-choice question-answering task.

We developed all models using the PyTorch Paszke et al. (2019) and PyTorch-Lightning Falcon (2019) deep learning frameworks. We also used the HuggingFace library for pre-trained models (ViLT, Dual Encoder, CLIP, and BLIP-2). We used an embedding size of $512$ for the Dual-Encoder and CLIP models, $768$ for the ViLT model, and $1408$ for the BLIP-2 model. We trained models using the AdamW optimizer with a weight decay regularization set to $0.01$ Loshchilov and Hutter (2017) and cosine annealing warm restarts with a cycle length ($T_0$): $\{2, 4, 6\}$, and cycle multiplier ($T_{mult}$): $2$. For the Dual Encoder, CLIP, ViLT, and BLIP-2 models doing detection we used a learning rate of $3e-5$, $3e^{-6}$, $3e^{-5}$, and $3e^{-6}$ respectively, and all models for VQA used a learning rate of $1e^{-5}$. We used a batch size of $32$ for all models except BLIP-2 where we used a batch size of $2$ due to the model being much larger. All models for detection were trained for 10 epochs on Refer360 and 25 epochs on CAESAR-PRO with a random seed of $33$; and all models for VQA were trained for 20 epochs with a random seed of $42$.

## 6 EXPERIMENTAL ANALYSIS

We have incorporated our proposed guided residual module MuRes into the existing state-of-the-art multimodal models, including CLIP Radford et al. (2021), DualEncoder Wu et al. (2019), ViLT Kim et al. (2021), BLIP-2 Li et al. (2023), and VisualBERT Li et al. (2019). We have evaluated these models and baselines multimodal models on Refer360 and CAESAR-PRO Islam et al. (2022c) datasets focusing on embodied referring expression comprehension (E-RFE) tasks. We have also evaluated these models on two more widely used datasets, ScienceQA Lu et al. (2022), and A-OKVQA Schwenk et al. (2022), to assess their performance on Visual Question Answering (VQA) tasks. We trained multiple variations of our proposed residual module MuRes, each differing in the type of residual representation of visual and language modalities. We examined four distinct variations:

- **Visual-Only Residual Representation MuRes(V)**: This variant leverages the projected visual representation as the query in the guided residual modules to extract the salient multimodal residual representations.

- **Language-Only Residual Representation MuRes(L)**: This variant utilizes the projected language representation as the query in the guided residual modules to extract the salient multimodal residual representations.

- **Visual and Language Residual Representation MuRes(V+L)**: This variant employs projected visual and language representations as the query to extract the salient multimodal residual representations.

- **Vanilla Models**: Following the original residual architecture He et al. (2016), this baseline directly summed visual and language representations to the projected representations without using any attention approach. We also evaluated several multimodal models in the vanilla mode without any residual connections.

### 6.1 EXPERIMENTAL EVALUATION ON EMBODIED REFERRING EXPRESSION COMPREHENSION TASK

We evaluated models on the Refer360 and CAESAR-PRO datasets for the embodied referring expression comprehension task. Following prior work on the embodied referring expression task Chen et al. (2021), we designed this task as an object bounding box detection task. All models were trained following a similar setup outlined in Section 5 (Training Model). We have reported Top-1 accuracy for the VQA tasks. The experimental results are presented in Table 3.

**Results and Discussion:** The experimental results in Table 3 indicate that augmenting existing multimodal models with the proposed multimodal guided residual module MuRes enhances embodied referring expression comprehension task performance on both the Refer360 and CAESAR-PRO datasets. More specifically, the results indicate that including **visual** reinforced representations enhances task performance. For example, augmenting MuRes into CLIPRadford et al. (2021) model

Table 3: Comparison of VL models performance on the embodied referring expression comprehension task, designed as bounding box detection. The results suggest that our multimodal guided residual module, MuRes, enhances the performance of most baseline multimodal models on the Refer360 and CAESAR-PRO datasets. Best performance numbers in **bold** face. (V: Visual, L: Language)

| Refer360 Dataset | | | | | | | | | | |
|---|---|---|---|---|---|---|---|---|---|---|
| Models | Without Residual | | Vanilla Residual | | MuRes(V) | | MuRes(L) | | MuRes(V+L) | |
| | IOU-25 | IOU-50 | IOU-25 | IOU-50 | IOU-25 | IOU-50 | IOU-25 | IOU-50 | IOU-25 | IOU-50 |
| CLIP | 25.80 | 7.67 | 27.22 | 8.35 | **29.20** | **9.15** | 28.30 | 7.50 | 26.65 | 7.27 |
| ViLT | 36.53 | 14.03 | 35.34 | 14.37 | - | - | - | - | **37.05** | **14.66** |
| BLIP-2 | **29.42** | 7.54 | 27.66 | 7.31 | 25.45 | 7.71 | 26.81 | **7.94** | 16.44 | 3.80 |
| Dual-Encoder | 31.08 | 9.83 | 30.17 | 8.98 | **31.36** | 8.92 | 29.43 | 9.03 | 31.08 | **10.68** |
| CAESAR-PRO Dataset Islam et al. (2022c) | | | | | | | | | | |
| Models | Without Residual | | Vanilla Residual | | MuRes(V) | | MuRes(L) | | MuRes(V+L) | |
| | IOU-25 | IOU-50 | IOU-25 | IOU-50 | IOU-25 | IOU-50 | IOU-25 | IOU-50 | IOU-25 | IOU-50 |
| CLIP | 37.92 | 9.82 | 39.43 | 10.83 | **42.91** | **11.91** | 39.56 | 10.85 | 39.06 | 10.46 |
| ViLT | 27.96 | **8.73** | 25.67 | 8.06 | - | - | - | - | **28.52** | 8.04 |
| Dual-Encoder | 42.52 | **12.14** | **42.61** | 11.61 | 36.72 | 8.51 | 37.97 | 10.32 | 37.72 | 11.50 |

and reinforcing visual representation improved object bounding detection task performance on our Refer360 dataset from 25.80% to 29.20% for IOU-25. Similarly, MuRes helps CLIPRadford et al. (2021) model enhance object bounding detection task performance on CAESAR-PRO Islam et al. (2022c) dataset from 37.92% to 42.91% for IOU-25. This performance improvement underscores the importance of visual cues in object grounding and suggests that reinforcing visual representation can lead to better performance.

Although the vanilla residual connection offers some performance improvement over models without any residual connection-based fusion, the gains are modest compared to those achieved with MuRes. The key distinction lies in MuRes's selective reinforcement of the most salient aspects of the visual-language representation, acting as an information bottleneck to extract only the relevant information. This targeted approach contrasts with vanilla residual connections, which indiscriminately reinforce the entire representation. These insights align with the findings from prior works on the information bottleneck Islam et al. (2023); Wang et al. (2022); Tishby and Zaslavsky (2015); Shwartz-Ziv and Tishby (2017); Tishby et al. (2000); Sun et al. (2022); Alemi et al. (2016); Träuble et al. (2022); Islam et al. (2024b). In the literature, it has been shown that information bottleneck helps the model to extract the relevant information and thus improve downstream task performance. Thus, the design choice of residual representation incorporation is pivotal in refining multimodal representation and, consequently, downstream task performance.

The experimental results further suggest that the specific modality being reinforced can influence performance improvements. For example, reinforcing the visual modality with MuRes boosts the CLIP model's performance for the object bounding box detection task from 25.80% to 29.20% for IOU-25. Conversely, emphasizing the language modality results in a slightly lower enhancement, with performance increasing to 28.30%. This variance suggests that the object grounding task is predominantly reliant on visual information.Thus, the choice of modality for reinforcement should be carefully considered based on the downstream task.

## 6.2 EXPERIMENTAL EVALUATION ON VISUAL QUESTION-ANSWERING TASK

We have evaluated the models on the ScienceQA Lu et al. (2022) and A-OKVQA Schwenk et al. (2022) datasets for the VQA task. Following the evaluation protocols in these benchmark datasets, we have evaluated the models on multiple-choice QA tasks. Similar to the previous tasks, we have incorporated different variations of our multimodal guided residual module MuRes in CLIP Radford et al. (2021), ViLT Kim et al. (2021), and VisualBERT Li et al. (2019) models. These variations are MuRes(V), MuRes(L), MuRes(V+L), and Vanilla Multimodal Models without residual connection for multimodal fusion. As ViLT is a monolithic model and provides combined visual-language representations, we split the output representation of the VILT model into separate representations for the text and image inputs based on the length of the text determined by the attention mask. All models were trained following the similar setup outlined in Section 5 (Training Model). We reported Accuracy for ScienceQA dataset and Multiple Choice (MC) based evaluation metric Schwenk et al. (2022) for AOK-VQA dataset. The experimental results are presented in Table 4.

Table 4: Comparison of VL models performance on the visual question-answering task. The results suggest that our multimodal guided residual module, MuRes, enhances the performance of the multimodal models on the ScienceQA and A-OKVQA datasets. Best performance numbers in **bold** face. (V: Visual, L: Language)

| ScienceQA Dataset Lu et al. (2022) | | | | | |
|---|---|---|---|---|---|
| Models | Without Residual | With Residual | MuRes(V) | MuRes(L) | MuRes(V+L) |
| CLIP | 21.31 | 33.36 | 40.75 | 31.33 | **51.85** |
| ViLT | 44.52 | 47.05 | 42.78 | 42.58 | **49.33** |
| VisualBERT | 34.95 | 36.63 | 37.13 | 37.63 | **39.03** |
| Dual-Encoder | 24.79 | 35.55 | 37.13 | 31.93 | **43.57** |
| A-OKVQA Dataset Schwenk et al. (2022) | | | | | |
| Models | Without Residual | With Residual | MuRes(V) | MuRes(L) | MuRes(V+L) |
| CLIP | 29.41 | **32.78** | **32.78** | 30.42 | 32.47 |
| ViLT | 31.61 | 31.21 | 32.19 | 31.48 | **32.53** |
| VisualBERT | 29.88 | 32.47 | 30.72 | 31.15 | **32.62** |
| Dual-Encoder | 32.64 | 33.45 | 32.89 | 31.72 | **35.02** |

**Results and Discussion:** The experimental results in Table 4 suggest that incorporating our multimodal guided residual module, MuRes, into multimodal models demonstrates consistent performance improvement across all variations evaluated compared to those without residual connections. Specifically, the inclusion of both visual and linguistic modalities (MuRes(V+L)) consistently yields the highest improvements. For example, in the ScienceQA dataset, CLIP model with MuRes VQA task accuracy increases from 21.31% to 51.85%. This performance improvement attributed to the information bottleneck in MuRes effectively extracts the salient representation from visual and language modalities, leading to more accurate answers.

The gains from visual-only (MuRes (V)) and language-only (MuRes (L)) reinforcements underscore the importance of modality-specific enhancements, with visual reinforcements being particularly impactful in the VisualBERT model on the ScienceQA dataset, improved its performance from 34.95% to 37.13% using visual reinforcement and 37.63% using language reinforcement. These insights suggest that strategically leveraging multimodal guided residuals can significantly refine model performance in VQA tasks.

## 7 CONCLUSION

In this paper, we have introduced a diverse dataset of multimodal interactions, Refer360, as well as presented a novel model, MuRes, to extract modality-specific salient representations. To comprehensively study embodied referring expressions in real-world settings, as our first contribution, we have curated a diverse dataset, Refer360, from various environments. We collected multimodal sensor data—exo visual view, ego visual view, depth, infrared, 3D skeletal data, audio, and robot camera motion—to capture unconstrained human interactions from multiple verbal and visual viewpoints. Consequently, Refer360 is the first embodied referring expression comprehension dataset curated with such diverse sensor data, which facilitates the study of embodied referring expressions. Additionally, we have conducted extensive experimental analyses, demonstrating that existing multimodal models cannot effectively understand embodied referring expressions in real-world settings. The primary reason for this discrepancy in performance is a failure to bridge the gap between general pre-trained frozen visual-language representations with salient modality-specific cues. To address this issue, as our second contribution, we have presented a multimodal guided residual module, MuRes. This module acts as a bottleneck to extract salient modality-specific representations, which are then integrated with the pre-trained representations. Our extensive quantitative and qualitative experiments suggest that incorporating MuRes into existing multimodal models improves downstream task performance on four datasets comprising embodied referring expression understanding and visual question answering. Our comprehensive multimodal dataset (Refer360), proposed multimodal guided residual module (MuRes), and findings from our experimental analyses show promising directions for research into embodied referring expression comprehension.

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
