# Technical Appendix

# Embodied Referring Expression Comprehension Through Multimodal Residual Learning

## A    Resources

- **Refer360 dataset (processed)** (49.71 **GB**):
  https://bit.ly/refer360_dataset_processed

- **Refer360 dataset (raw)** (2572.36 **GB**):
  https://bit.ly/refer360_dataset_raw

- **Source code of Refer360 data collection system:**
  https://bit.ly/source_code_data_collection_system

- **Source code of MuRes and baseline models** (8.8 **MB**):
  https://bit.ly/source_code_MuGuRu_and_baseline_models

- **Trained model checkpoints of CLIP with MuRes for embodied referring expression task** (5.4 **GB**):
  https://bit.ly/model_checkpoints

- **Docker for training models** (8.59 **GB**): We built a docker to facilitate easy reproducing of our experimental settings and training environment. We cannot currently share the docker hub link to maintain anonymity. We plan to share that docker link upon publication of the paper. For this reason, we are sharing the singularity container built from the same docker we used for our experimentation: https://bit.ly/multimodal-docker

## B    Additional Experimental Results: Quantitative Analysis

We have performed a quantitative evaluation of the models by applying the ScienceQA (20) and A-OKVQA (30) datasets for the visual-question answering tasks. We have analyzed the response of VisualBERT with different variations of our proposed model, (MuRes), on multiple-choice question-answering tasks. The responses from VisualBERT model variations are similar to the variation presented in Table 1 from the manuscript (i.e., without residual, MuRes (V), MuRes (L), and MuRes (V+L)).

**Discussion:**   The model responses are presented in Fig. 1. These results suggest that augmenting the VisualBERT model with MuRes improves responses for the visual question-answering task. For instance, in Fig. 1 (a) [Q-A1], the VisualBERT model's response to the question *"Which continent is highlighted?"* alongside an image of a map shows that enhancing visual representations through MuRes yields the correct answer (*"Europe"*). However, enhancing only the language representations through MuRes leads to an incorrect answer (*"Asia"*). This question necessitates a thorough understanding of the spatial location of the highlighted region (*"Europe"*) on the map, explaining why reinforcing the visual representations aids in improving the response. Conversely, in Fig.1 (a) [Q-A3], enhancing either visual or language representations does not yield the most accurate answer (*"Transparent"*) for the question: *"Which property do these three objects have in common?"*. Although the responses with either Vision or Language in Fig.1 (a) [Q-A3] are not entirely inaccurate, as the objects are somewhat shiny, only yhe model with both visual and language representations reinforced correctly answers "Transparent". Therefore, identifying which modalities to reinforce thorough MuRes is a critical aspect of enhancing the model's responses.

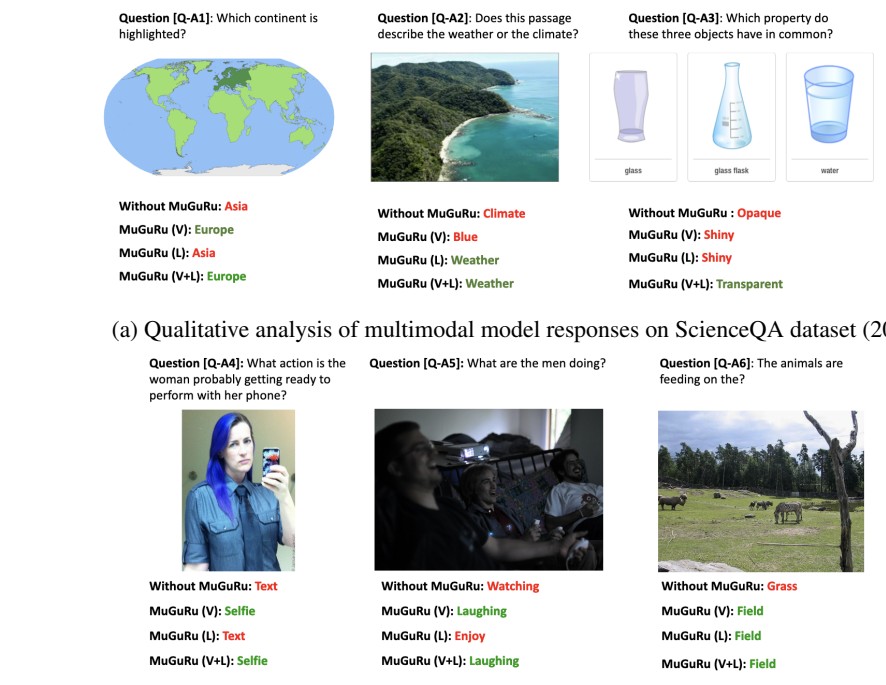

(a) Qualitative analysis of multimodal model responses on ScienceQA dataset (20).

(b) Qualitative analysis of multimodal model responses on A-OKVQA dataset (30).

Figure 1: We evaluated VisualBERT with different variations of the multimodal guided residual (MuRes) on the ScienceQA and A-OKVQA datasets. The results suggest that incorporating MuRes using guided residual visual and/or language representations improves the performance of the visual question-answering task on both datasets.

## C   DATA COLLECTION

### C.1   DATA COLLECTION SYSTEM

Our data collection system integrates an Azure Kinect DK (1) a and a Pupil Smart Glass, also known as the Pupil Invisible Eye Tracker (4). The Azure Kinect DK was mounted on an Ohmni Telepresence robot (3), and the participants wore the Pupil Smart Glass to facilitate data collection in real-world scenarios. An Alienware m15 R4 laptop powered by an i7-10870H RTX processor served as the high-performance computing backbone. A Python-based application was developed to facilitate coordination and synchronization among all system components. This application ensured seamless operation and synchronized data collection from multiple sensors.

### C.1.1   SENSOR SPECIFICATIONS

Azure Kinect provides a multitude of sensory data, including visual, depth, infrared (IR), skeletal tracking, and inertial measurement unit (IMU) data. In addition, pupil glass offers visual (RGB), IR, gaze tracking, and gesture recognition capabilities. The Pupil Invisible Eye Tracker is a state-of-the-art device with a range of features designed to capture precise and accurate eye-tracking data. The participants in our study were equipped with the Pupil Smart Glass and an Android smartphone, which recorded their eye-tracking data. The data is subsequently transmitted to the Pupil Cloud via the Pupil Invisible Android application. This seamless hardware and software integration ensures efficient and reliable data collection and transmission. The specifications of the Azure Kinect DK and Pupil Eye Tracker sensors are listed in Table 3 and 4.

Table 1: Comparison of the embodied referring expression datasets. Most of the existing VQA and EQA datasets do not contain nonverbal gestures (NV), multiple verbal (V) perspectives (MP), and outdoor scene data samples. ‡Embodied (E) interactions refer to humans interacting using multimodal expressions. †Embodied interactions refer to an agent navigating in an environment. ⋆Sythetic Environment.

| Datasets | V | NV | E | MV | Views | |
|---|---|---|---|---|---|---|
| | | | | | Exo | Ego |
| PointAt (29) | ✗ | ✓ | ✓ | ✗ | ✓ | ✗ |
| ReferAt (28) | ✓ | ✓ | ✓ | ✗ | ✓ | ✗ |
| IPO (31) | ✗ | ✓ | ✓ | ✗ | ✓ | ✗ |
| IMHF (32) | ✗ | ✓ | ✓ | ✗ | ✓ | ✗ |
| RefIt (16) | ✓ | ✗ | ✗ | ✗ | ✓ | ✗ |
| RefCOCO (36) | ✓ | ✗ | ✗ | ✗ | ✓ | ✗ |
| RefCOCO+ (36) | ✓ | ✗ | ✗ | ✗ | ✓ | ✗ |
| RefCOCOg (22) | ✓ | ✗ | ✗ | ✗ | ✓ | ✗ |
| Flickr30k (26) | ✓ | ✗ | ✗ | ✗ | ✓ | ✗ |
| GuessWhat? (9) | ✓ | ✗ | ✗ | ✗ | ✓ | ✗ |
| Cops-Ref (7) | ✓ | ✗ | ✗ | ✗ | ✓ | ✗ |
| CLEVR-Ref+ (19) | ✓ | ✗ | ✗ | ✗ | ✓ | ✗ |
| DAQUAR (21) | ✓ | ✗ | ✗ | ✗ | ✓ | ✗ |
| FM-IQA (10) | ✓ | ✗ | ✗ | ✗ | ✓ | ✗ |
| Visual Madlibs (35) | ✓ | ✗ | ✗ | ✗ | ✓ | ✗ |
| Visual Genome (17) | ✓ | ✗ | ✗ | ✗ | ✓ | ✗ |
| DVQA (15) | ✓ | ✗ | ✗ | ✗ | ✓ | ✗ |
| VQA (COCO) (5) | ✓ | ✗ | ✗ | ✗ | ✓ | ✗ |
| VQA (Abs.) (5) | ✓ | ✗ | ✗ | ✗ | ✓ | ✗ |
| Visual 7W (37) | ✓ | ✗ | ✗ | ✗ | ✓ | ✗ |
| KB-VQA (34) | ✓ | ✗ | ✗ | ✗ | ✓ | ✗ |
| FBQA (33) | ✓ | ✗ | ✗ | ✗ | ✓ | ✗ |
| VQA-MED (12) | ✓ | ✗ | ✗ | ✗ | ✓ | ✗ |
| DocVQA (23) | ✓ | ✗ | ✗ | ✗ | ✓ | ✗ |
| YouRefIt (6) | ✓ | ✓ | ✓ | ✗ | ✓ | ✗ |
| GRiD-3D (18) | ✓ | ✗ | ✗ | ✗ | ✓ | ✗ |
| EQA † (8) | ✓ | ✗ | ✗ | ✗ | ✓ | ✗ |
| MT-EQA † (8) | ✓ | ✗ | ✗ | ✗ | ✓ | ✗ |
| CAESAR-L (14) | ✓ | ✓ | ✓ | ✓ | ✓ | ✓ |
| CAESAR-XL (14) | ✓ | ✓ | ✓ | ✓ | ✓ | ✓ |
| EQA-MX (13) | ✓ | ✓ | ✓ | ✓ | ✓ | ✓ |
| Refer360 | ✓ | ✓ | ✓ | ✓ | ✓ | ✓ |

### C.1.2 DATA COLLECTION APPLICATION

We developed a Python application to coordinate and synchronize the various components of our data collection system. This application played a central role, ensuring seamless integration and synchronized data capture from multiple sensors. We collected camera video feeds, time-series data from the inertial measurement unit (IMU) and skeleton joint positions, and session metadata using this system. We utilized the pyKinectAzure (11) python library to interface with the Azure Kinect SDK sensor, while the Pupil Labs' Real-time Python API (27) facilitated communication with the Pupil Eye camera. Participants stood before the Ohmin robot, issuing verbal commands and nonverbal gestures to reference physical objects. An RGB camera on the Azure Kinect device continuously captured visual data, providing a third-person view of the participants' referencing gestures. Additionally, the Kinect's depth and infrared sensors recorded supplementary data streams, enriching the external perspective of the interactions. The system also leveraged the Kinect's infrared sensor to collect infrared data and the Azure Kinect Body Tracking SDK (24) to capture the 3D coordinates and orientations of 32 skeletal joints. Simultaneously, the Kinect's microphone recorded the participants' verbal instructions. Complementing this external viewpoint, the Pupil Invisible Eye Tracker provided an egocentric visual stream from the participants' perspectives. Combining

Table 2: Comparison of the embodied referring expression datasets. Most of the existing VQA and EQA datasets do not contain nonverbal gestures (NV), multiple verbal (V) perspectives (MP), and outdoor scene data samples. ‡Embodied (E) interactions refer to humans interacting using multimodal expressions. †Embodied interactions refer to an agent navigating in an environment. ⋆Sythetic Environment.

| Datasets | No. of Images | No. of Samples | Object Categories | Avg. Words* |
|---|---|---|---|---|
| PointAt (29) | 220 | 220 | 28 | - |
| ReferAt (28) | 242 | 242 | 28 | - |
| IPO (31) | 278 | 278 | 10 | - |
| IMHF (32) | 1716 | 1716 | 28 | - |
| RefIt (16) | 19,894 | 130,525 | 238 | 3.61 |
| RefCOCO (36) | 19,994 | 142,209 | 80 | 3.61 |
| RefCOCO+ (36) | 19,992 | 141,564 | 80 | 3.53 |
| RefCOCOg (22) | 26,711 | 104,560 | 80 | 8.43 |
| Flickr30k (26) | 31,783 | 158,280 | 44,518 | - |
| GuessWhat? (9) | 66,537 | 155,280 | - | - |
| Cops-Ref (7) | 75,299 | 148,712 | 508 | 14.40 |
| CLEVR-Ref+ (19) | 99,992 | 998,743 | 3 | 22.40 |
| DAQUAR (21) | 1449 | 124,68 | 37 | 11.5 |
| FM-IQA (10) | 157,392 | 316,193 | - | 7.38 |
| Visual Madlibs (35) | 107,38 | 360,001 | - | 6.9 |
| Visual Genome (17) | 108,000 | 1,445,332 | 37 | 5.7 |
| DVQA (15) | 300,000 | 3,487,194 | - | - |
| VQA (COCO) (5) | 204,721 | 614,163 | 80 | 6.2 |
| VQA (Abs.) (5) | 50,000 | 150,000 | 100 | 6.2 |
| Visual 7W (37) | 47,300 | 327,939 | 36,579 | 6.9 |
| KB-VQA (34) | 700 | 5826 | 23 | 6.8 |
| FBQA (33) | 2190 | 5826 | 32 | 9.5 |
| VQA-MED (12) | 2866 | 6413 | - | - |
| DocVQA (23) | 12,767 | 50,000 | - | - |
| YouRefIt (6) | 497,348 | 4,195 | 395 | 3.73 |
| GRiD-3D (18) | 8,000 | 445,000 | 28 | - |
| EQA † (8) | 5,000 | 5,000 | 50 | - |
| MT-EQA † (8) | 19,287 | 19,287 | 61 | - |
| CAESAR-L (14) | 11,617,626 | 124,412 | 61 | 5.56 |
| CAESAR-XL (14) | 841,620 | 1,367,305 | 80 | 5.32 |
| EQA-MX (13) | 750,849 | 8,243,893 | 52 | 11.45 |
| Refer360 | 2,472,939 | 28,736 | 75 | 11.45 |

these exocentric and egocentric data sources gave the system a comprehensive understanding of human-robot interactions.

We stored the Azure Kinect recordings and the corresponding keystroke event times locally as MP4 and JSON files, respectively. For the Pupil eye tracker, the recordings of the participants' ego view and keystroke events were saved in the Pupil Cloud using the Pupil Lab Android app and Pupil API, respectively.

### C.1.3 Time-based Synchronization

One of the significant challenges we faced was synchronizing the various data streams captured by different devices. To address this, we implemented a time-based synchronization method that recorded the UNIX timestamps of different data capture events and data streams, enabling synchronization during post-processing. This synchronization is crucial for aligning the data streams captured from different devices. Our approach involved recording the timestamp at the start and end of each interaction and the timestamp of the event when the participant pointed to an object (i.e., canonical events). This was achieved using our Python-based system, which is operated by individuals recording the data collection sessions. We utilized different keystrokes on a standard keyboard

grey ceramic bowl, foam miniature football, wireless computer mouse, wooden box, blue cupholders, plastic water bottle, keyboard, green plastic cup, white plastic basket, basketball, white plastic cup, flower vase, clorox wipe container, paper towel roll, mountain dew bottle, picture frame, TV remote, grey plastic basket, black metal water bottle, coffee cup with lid, transformers robot, pepsi bottle, egg carton, TV screen, blue plastic box, pringles box, grey dustbin, light green open plastic box with handle, tripod, white three-level plastic box, cardboard box, sunglasses, yellow lego box, mouthwash, pink plastic cup, white tumbler, white desk fan, blue plastic container with lid and handle, salsa jar, nutella jar, pink dustbin, black kickball, table tennis ball container, blue plastic water bottle, black desk clock, screwdriver, blue magazine, shoe rack, bicycle, pupil labs glasses box, microwave, frying pan, blue couch, wooden chair, white rope, kitchen sink, white fridge, iron stand, allen wrench set, white trash can, black dresser, light stand, desk lamp, black office chair, silver rice cooker, black standing fan, wooden table, white pillow, white air conditioning unit, grey sweatshirt, banana, grey laundry drying rack, grey apartment mailboxes, white fence, surge protector.

Figure 2: Objects in Refer360 Dataset

Table 3: Azure Kinect DK Sensor Specifications

| Sensor | Specification |
| --- | --- |
| RGB Camera | Highest Resolution: $3840 \times 2160$ px @ 30 fps |
| Depth Camera | Method: Time-of-Flight, Highest Resolution: $640 \times 576$ px @ 30 fps |
| Motion Sensor | LSM6DSMUS IMU (accelerometer & gyroscope), Sampling Rate: 1.6 Hz |
| Microphone | USB audio 2.0, Channels: 7, Sensitivity: $-22$ dBFS (94 dB SPL, 1 kHz), SNR: $> 65$ dB, Acoustic Overload Point: 116 dB |

Table 4: Pupil Invisible Eye Tracker Specifications

| Sensor | Specification |
| --- | --- |
| Eye Cameras | 200 Hz @ $192 \times 192$ px, IR illumination |
| Scene Camera | 30 Hz @ $1088 \times 1080$ px, $82° \times 82°$ FOV |

to denote different events. The "Space" key was pressed at the start and end of an interaction, while the "G" key was pressed to identify the canonical event of an interaction. The canonical event indicates when the participant points to an object using gaze or pointing gestures. Specifically, the "G" keystroke event time was used to identify the canonical frame, i.e., the frame where the participant actually pointed to an object. When the participant used cues other than pointing, such as gaze, the "G" key was pressed when the gaze event occurred. The "Space" keystroke event time was used to identify the start and end of an interaction, thereby facilitating the segmentation of interactions. The "Q" key was used to terminate a session.

The corresponding UNIX timestamp for these keystroke events was recorded for both the Azure Kinect and Pupil Lab Eye Tracker. This enabled us to synchronize the data streams from these two devices during post-processing. Though the time-based synchronization method is utilized to synchronize between the Azure Kinect Sensor and Pupil Eye Tracker, it is designed to be extensible. For example, our system can be expanded to incorporate multiple Azure Kinect devices to capture multiple views of the participant during interaction rather than just the ego and exo views.

### C.1.4 Data Collection Environment

The Refer360 dataset aims to study real-world human-robot interactions in which a human provides object-referencing instructions to robots across diverse environments, ranging from controlled laboratory setups to outdoor locations. Refer360 contains embodied interaction data from lab and outside-lab environments. The outside lab refers to settings outside controlled lab settings, such as homes, outdoor locations, etc. While choosing objects, we prioritize those usually available in

these environments. Our dataset contains 75 objects from the aforementioned environments, and a complete list of objects is given in Fig. 2.

## C.2 DATA COLLECTION PROTOCOL AND PROCEDURE

The data collection process began with a comprehensive introduction to the system, the purpose of the dataset, and the protocol to be followed during collection. Before participating in the data collection sessions, subjects completed a demographic survey.

Each session involved subjects providing embodied instructions that referenced objects in their surroundings, using both language and nonverbal gestures (gaze and pointing gestures). The ultimate goal of this dataset is to enhance social robots' ability to interpret object referencing instructions accurately. This involves uniquely identifying the object, which requires extracting the object's location and other attributes from the instruction. This task is challenging as humans often use diverse formats when providing verbal instructions, and these instructions may sometimes lack the necessary features for object identification. Incorporating nonverbal cues, such as pointing or referencing the object in relation to another object, can significantly improve the efficiency of interpreting object referencing instructions. Furthermore, object referencing instructions can be given from multiple perspectives, such as the subject's or the robot's perspective, which must be resolved for accurate object comprehension.

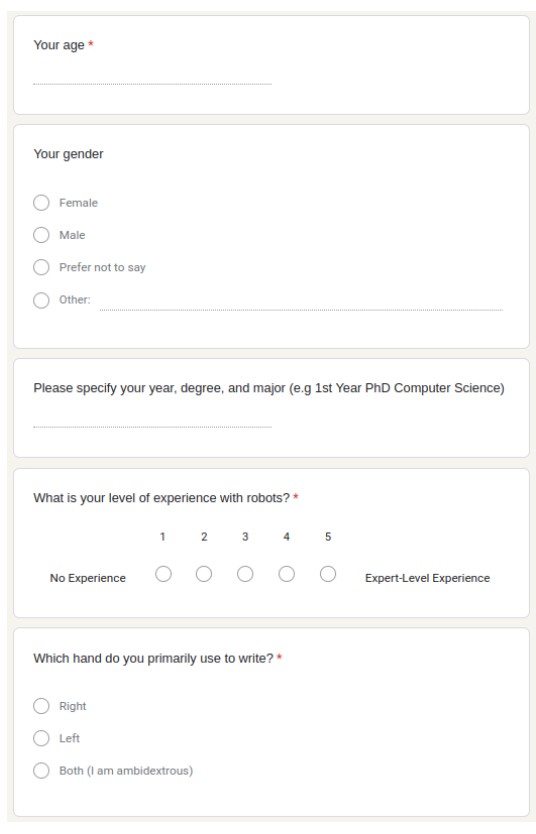

Figure 3: Demographic Survey

The participants were given the flexibility to choose any perspective (subject, robot, or neutral) when providing instructions. This approach allowed us to diversify our dataset by including object-referencing instructions with varied spatial referencing and perspectives. For instance, an object could be referenced in relation to another object, such as "The black box on top of the brown table." The object reference in the verbal instruction could be from the subject's perspective, e.g., "The couch to my right," or it could be from the robot's perspective, e.g., "The lamp to your left."

We had two distinct data collection conditions: constrained and unconstrained. In the constrained condition, subjects were briefed on the format of instructions and how they could employ various modalities (verbal and nonverbal) to make the interaction as natural as possible. We also suggested that participants use both verbal and nonverbal gestures to describe an object. In the unconstrained condition, we did not suggest whether to use verbal or nonverbal gestures to describe an object. We instructed the participant to describe an object to the robot.

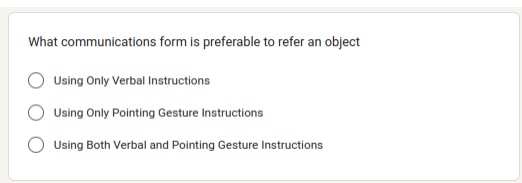

Figure 4: Post-Task Survey

This allowed us to capture natural human instincts when providing instructions. This approach also helped eliminate biases that might be introduced by pre-guidance on the format of the instructions, allowing subjects to be flexible in their instruction delivery.

Each subject participated in multiple sessions, each lasting approximately one hour. During each session, the subject performed several interactions. Using our data collection system, we recorded the subject's ego view, exo view, IMU, skeleton, and audio data stream for each session. Upon completion of the sessions, subjects were asked to complete a post-task survey and sign a consent form to permit the release of the dataset. The University's IRB approved the study. The demographic and post-task surveys are presented in Figure3 and 4.

### C.3 DATASET PROCESSING

The developed Python-based application generated an Azure Kinect video file in MP4 format for each session. The MP4 file contains three data streams from Azure Kinect's camera sensor: RGB, Depth, and Infrared. Separate JSON files contain the IMU and skeleton joints' time series data and relevant session metadata. We utilized the FFmpeg (2) library to extract the Kinect video streams into separate MP4 files and the recording audio as an MP3 file. The IMU time series was split into two different files for the accelerometer and gyroscope readings. For each session, the Pupil eye tracker also generated one video file in MP4 format and saved it to the pupil cloud.

The major challenge of data post-processing was segmenting the interactions and synchronizing the Azure Kinect and Pupil lab data. For the segmentation of each interaction from Azure Kinect data streams, we look into that interaction's start and end time. We also identify the canonical frames, i.e., frames where the subject points precisely to the object. We split each interaction and canonical frame using the FFmpeg library. Next, we searched the corresponding Pupil recording for the Azure Kinect recording from the pupil cloud using Python Pupil Cloud API. For this purpose, we used the recording-start timestamp saved in the metadata file to find the matching Pupil recording in the pupil cloud. After downloading the Pupil video, we employed the same procedure as Azure Kinect recording to split the interactions and canonical frames at the timestamps recorded during data collection. Finally, we utilized the OpenAI whisper (25) library to transcribe Kinect audio data to the corresponding text. Note that we manually verified the synchronization and segmentation with five human experts whom the IRB approved. Subsequently, the dataset underwent annotation by human annotators sourced from an external company specializing in data annotation services, ensuring accuracy and reliability.

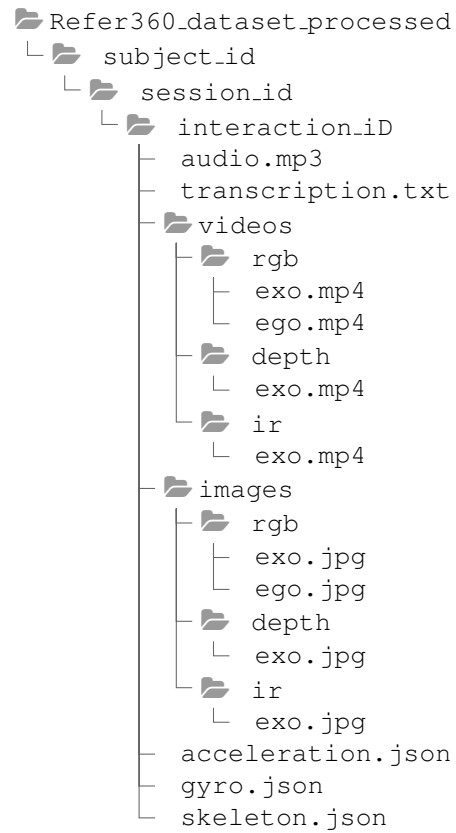

Figure 5: Refer360 dataset folder structure.

Our dataset contains several data collection sessions and after data post-processing results in each session's folder structure shown in Figure 5. Here, *transcription.txt* is the text transcription of audio.mp3. In the subfolders in *Videos* and *Frames*s, *exo.mp4* and *ego.mp4* refer to the videos from the Azure Kinect SDK camera and Pupil Eye Camera, respectively.