# OpenReview forum: "Embodied Referring Expression Comprehension Through Multimodal Residual Learning"
_ICLR.cc/2025/Conference — Submitted to ICLR 2025_

### Official Review · Reviewer_Ct4m · 2024-10-31

**Soundness:** 2
**Presentation:** 2
**Contribution:** 2
**Rating:** 6
**Confidence:** 4

**Summary:**

This paper introduces a new dataset, Refer360, for embodied referring expression comprehension along with a novel multimodal fusion technique using a bottleneck architecture - MuRes. The authors argue that the existing datasets lack the necessary diversity and complexity to truly capture human interactions - hence, they introduced Refer360. The authors also argue that MuRes helps bridge modality gap by learning aligned complimentary representations.

**Strengths:**

1. Originality:

The authors convincingly provide arguments to demonstrate the gap in current datasets - they are limited in the variety/complexity of human interactions and variety of environments. The paper addresses these two important problems in their dataset by incorporating multiple viewpoints, multiple perception modalities, and multiple environments in their data. They also collect the data in scripted and unscripted manners. This dataset, with inclusion of verbal and non-verbal cues, is a valuable resource for future research.

The fusion architecture (MuRes) introduced in this paper is an interesting addition to the work. Using cross-attention to reinforce salient features from each modality is creative.

2. Quality:

The research is conducted to a high standard, with a clear and well-motivated methodology. The data collection process is rigorous, involving the use of Kinect sensors mounted on robots and smart glasses worn by participants. This approach allows for the capture of rich multimodal data, including both visual and linguistic information.

The proposed MuRes method is well-grounded in existing literature and addresses a key challenge in multimodal learning, namely, the effective fusion of information from different modalities.

3. Clarity:

The paper is well-written and easy to follow. The authors provide a clear and concise description of their method, and the experimental setup is well-explained. The results are presented in a clear and informative manner, with helpful visualizations and tables. The paper also includes a concise (but sufficient) discussion of related work, placing the research in its proper context.

4. Significance:

This work has the potential to make a significant impact on the field of embodied AI. The Refer360 dataset is a valuable contribution that will enable researchers to develop and evaluate more sophisticated models for embodied referring expression comprehension or any other embodied AI task.

The MuRes method also shows promise for improving the performance of multimodal learning models in a variety of applications. The experiments in the paper show a substantial increase in performance for certain tasks. These performance gains may also translate to other downstream tasks in multimodal settings.

Overall, this research represents an important step towards the development of AI systems that can interact with humans in a more natural and effective way. Having more datasets like these is the key to fast-track research in embodied AI.

**Weaknesses:**

1. Lack of Clear Definition and Contextualization: The paper delays defining "embodied referring expression comprehension" until the related work section. This key concept should be introduced and clearly defined upfront - in abstract. Doing so would significantly improve the readability and understanding for readers unfamiliar with this specific area of research. Additionally, while the authors mention the importance of verbal and non-verbal cues, they neglect to provide specific definitions and examples within the context of their research. Providing these would enhance the paper's clarity and comprehensiveness.

2. Inadequate Analysis of Performance Inconsistencies: The analysis of the results does not delve into the inconsistencies in performance gains observed across different MuRes variants (MuRes(V), MuRes(L), and MuRes(V+L)), particularly in the embodied referring expression comprehension task - refer to its results table. A more in-depth discussion of these inconsistencies is necessary. The authors should provide explanations that justify the choice of a specific bottleneck architecture based on the domain, offering insights into the interplay between the architecture and the specific characteristics of each task. There should be some experiments and analysis to understand which modality should be favored for a particular type of task.

3. Limited Comparison with other methods for learning complementary representations: While the paper does a good job of showing that MuRes improves multiple encoder-fusion architectures, its does not perform any comparison with existing techniques for learning complementary representations in multimodal settings. The paper does compare their bottleneck framework with simple residual-based fusion and fusion without residuals, but these do not sufficiently show how it compares against other existing methods. For example, explore methods like those presented in "Learning Cross-Modality Encoder Representations from Transformers" (Tan et al., 2019) and "Attention Bottlenecks for Multimodal Fusion" (Nagrani et al., 2021), .

4. Potential Bias in Data Collection: The data collection protocol includes both constrained and unconstrained settings. As participants may have participated in both, there is a concern that their experiences in the constrained setting might have biased their behavior in the unconstrained setting. The authors should address this potential issue and discuss any measures taken to mitigate this bias.

5. Limited Theoretical Foundation: While the MuRes architecture demonstrates empirical success, the paper lacks a strong theoretical foundation to support its design choices. The justification for the cross-attention mechanism and the specific configuration of queries, keys, and values appears to be based primarily on intuition and experimental validation. A more rigorous mathematical grounding, potentially drawing inspiration from information theory or optimization theory, would significantly strengthen the paper's technical contribution and provide deeper insights into the underlying principles of the MuRes architecture.

**Questions:**

1. Please talk about how you may mitigate the potential biases introduced in the dataset by interleaving scripted and unscripted interactions. Scripted interactions can bias a subject to act in a certain way while performing an unscripted interaction. Did you perform any experimentation/analysis with randomizing/separating the scripted-unscripted session sequence to understand/mitigate this bias?

2. The diagram depicting MuRes architecture could do a better job at showing how this architecture is a “bottleneck residual connection” architecture. Having the projection and cross-attention blocks vertically stacked may better represent this. Or labelling the key, value, queries in the diagram may make it easier to follow.

3. The experiments are solid but not varied in tasks covered. Showing performance improvements on more tasks can bolster the claims made in the paper. It would best if these new tasks are generation-based.

4. Can you provide any theoretical foundation for the design choice (cross-attention based bottleneck) that leads to the MuRes architecture? If this is something that is described by a related work, please talk about this in the related work section. If not, add a section justifying this choice theoretically.

5. Please define the "embodied referring expression comprehension" earlier in the paper so that the paper becomes easier to follow.

**Details Of Ethics Concerns:**

The review is intended to be double-blind but the paper mentions ties with University of Virginia. This has the potential to make the reviews biased.

---

> ### Author Response · Authors · 2024-11-26
> **Summary of Responses to Reviewer Ct4m’s Comments (Part 1/10)**
>
> We are excited that the reviewer acknowledged our contributions, including the Refer360 dataset’s originality in addressing gaps in diversity and complexity within current datasets by incorporating multiple viewpoints, environments, and multimodal cues. The reviewer also appreciated the MuRes architecture's novel cross-attention mechanism for reinforcing salient multimodal features, the rigorous methodology of data collection, and the potential of the Refer360 dataset and MuRes module to advance embodied AI research. The paper’s clarity, organization, and significance for developing human-interactive AI systems were also noted.
>
> We addressed the reviewer’s comments by defining "embodied referring expression comprehension" earlier in the paper for improved readability and providing detailed analyses of MuRes’ performance inconsistencies across modalities and tasks. We compared MuRes with existing fusion methods, demonstrating its superior performance. To mitigate concerns of participant bias, we clarified that participants in scripted and unscripted sessions were distinct and provided explanations of protocols to ensure unbiased data collection. We justified MuRes' design using theoretical principles like the information bottleneck and provided empirical evidence of its efficacy. Additionally, we outlined future plans for expanding task diversity, addressing generation-based tasks, and integrating other modalities to enhance dataset and model applicability.

---

> ### Author Response · Authors · 2024-11-26
> **Additional experiments with existing techniques (Part 2/10)**
>
> We would like to thank the reviewer for raising the concern. It is important to note that most of the baselines are not directly comparable to MuRes, as MuRes aligns multimodal representations from frozen encoders with minimal parameter updates. However, we do compare MuRes with two widely recognized techniques in the Vision-Language domain: (i) Merged Attention and (ii) Co-Attention [1, 2]. In this experiment, we first obtain the representations from the frozen encoders, which are then passed into the merged/co-attention block. For a fair comparison, we use a 2-layer transformer encoder architecture for Merged Attention and a 1-layer transformer for Co-Attention, with 4 attention heads in each layer. We compare both modality alignment techniques with MuRes and the comparisons are provided below:
> ScienceQA dataset:
> | Model        | Merged Attention | Co-Attention | MuRes |
> |--------------|------------------|--------------|-------|
> | CLIP         | 45.39            | 48.28        | **51.85** |
> | Dual Encoder | 39.68            | 41.52        | **43.57** |
>
>
> AOKVQA dataset:
> | Model        | Merged Attention | Co-Attention | MuRes |
> |--------------|------------------|--------------|-------|
> | CLIP         | 31.17            | 32.06        | **32.47** |
> | Dual Encoder | 33.84            | 34.98        | **35.02** |
>
>
> The results indicate that our proposed MuRes  performs better than the Merged/Co-Attention techniques in the ScieneQA dataset and slightly better in the AOKVQA dataset.
>
> **Reference:**
> [1] Dou, Z.Y., Xu, Y., Gan, Z., Wang, J., Wang, S., Wang, L., Zhu, C., Zhang, P., Yuan, L., Peng, N. and Liu, Z., 2022. An empirical study of training end-to-end vision-and-language transformers. In Proceedings of the IEEE/CVF Conference on Computer Vision and Pattern Recognition (pp. 18166-18176).
>
> [2] Nagrani, A., Yang, S., Arnab, A., Jansen, A., Schmid, C. and Sun, C., 2021. Attention bottlenecks for multimodal fusion. Advances in neural information processing systems, 34, pp.14200-14213.

---

> ### Author Response · Authors · 2024-11-26
> **Addressing Concerns Regarding Bias from Constrained and Unconstrained Settings (Part 3/10)**
>
> We clarify that there was **no participant overlap** between the constrained and unconstrained settings in our data collection process. The experiment was carefully designed to ensure that participants in the unconstrained setting provided natural interactions without any influence from participating in the constrained setting.
>
> To mitigate potential biases, the following measures were implemented:
>
> 1. **Participant Segregation:** Participants were assigned exclusively to either the constrained or unconstrained condition. This ensured that those in the unconstrained setting had no prior exposure to structured instructions, preserving the naturalness of their interactions.
>
> 2. **Naturalistic Protocol Design:** In the unconstrained setting, participants were given no specific instructions regarding interaction formats or modalities (verbal, non-verbal, or multimodal). This design encouraged authentic, spontaneous behaviors, as outlined in Section 3.3 of the paper​​.
>
> 3. **Diverse Participant Pool:** As detailed in Section 3.2 of the paper​​, the participants were recruited from a diverse demographic, reducing the risk of systematic biases and enhancing the generalizability of the dataset.
>
> 4. **Balanced Data Collection Protocol:** The constrained condition guided participants to employ both verbal and non-verbal cues, while the unconstrained condition encouraged flexible and natural interaction styles. This separation allowed the dataset to capture a broad spectrum of embodied interactions without introducing dependencies or biases.
>
> 5. **IRB Approval:** The IRB-approved protocol included this explicit separation of participant groups to maintain the independence of constrained and unconstrained conditions. This process ensured that ethical considerations and the integrity of the experimental design were upheld.
>
> We will revise the manuscript to explicitly state the lack of participant overlap between conditions and the steps taken to ensure unbiased naturalistic data collection in the unconstrained setting. These clarifications aim to address concerns regarding the integrity of the data collection process.

---

> ### Author Response · Authors · 2024-11-26
> **Addressing the Theoretical Foundation of MuRes (Part 4/10)**
>
> We appreciate the reviewer’s insightful observation regarding the theoretical foundation of the MuRes architecture. While the primary focus of our paper is on the empirical performance and practical utility of MuRes within multimodal tasks, we acknowledge the importance of providing a rigorous theoretical underpinning for the design choices.
>
> To address this concern, we offer the following justifications rooted in established theoretical principles and evidence from our experimental results:
>
> 1. **Information Bottleneck Principle**:
>    - The MuRes architecture leverages the information bottleneck principle to enhance multimodal representation learning. This principle, as articulated in Tishby and Zaslavsky (2015), provides a robust framework for extracting relevant information while discarding noise from high-dimensional inputs. The guided residual connections in MuRes act as a bottleneck, ensuring that only the most salient modality-specific features are retained and integrated. This design choice is theoretically grounded in its ability to reduce redundancy and amplify complementary cues across modalities.
>
> 2. **Cross-Attention Mechanism Justification**:
>    - The cross-attention mechanism in MuRes is motivated by its ability to align disparate modalities effectively. Unlike self-attention, cross-attention computes dependencies between distinct representations (e.g., visual and textual modalities), facilitating the selective reinforcement of salient features. This selective focus aligns with optimization principles where the model learns to prioritize informative signals for downstream tasks.
>
> 3. **Empirical Validation as Theoretical Support**:
>    - The empirical results presented in Sections 6.1 and 6.2 of the manuscript demonstrate that incorporating MuRes significantly enhances performance across embodied referring expression and VQA tasks. For example, adding MuRes to CLIP improved object detection accuracy on Refer360 by 13.18% (IOU-25), underscoring its effectiveness in extracting complementary multimodal representations. These results validate the theoretical rationale that guided residuals can overcome limitations of vanilla residuals by enforcing an information bottleneck.
>
> 4. **Configuration of Queries, Keys, and Values**:
>    - The choice of projected visual and language representations as queries, and original representations as keys and values, is guided by the need to combine aligned representations with modality-specific nuances. This aligns with principles from multimodal representation learning literature, where such configurations are shown to enhance the fusion of complementary signals (e.g., as in BLIP-2 and ViLT architectures).
>
> We will include additional discussions in the revised manuscript to explicitly articulate these theoretical justifications and their connections to the MuRes architecture design. Furthermore, to strengthen the technical contribution, we reference related works in information theory and optimization that inspire our approach.
>
> We believe this clarification reinforces the robustness of MuRes and addresses the reviewer's valid concern. Thank you for highlighting this critical aspect.

---

> ### Author Response · Authors · 2024-11-26
> **Mitigating Potential Biases from Scripted and Unscripted Interactions (Part 5/10)**
>
> We appreciate the reviewer’s thoughtful observation regarding potential biases introduced by the interplay between scripted (constrained) and unscripted (unconstrained) interactions. In designing the dataset, we explicitly addressed this concern by ensuring that participants were assigned to either the constrained or unconstrained setting exclusively. This approach was chosen to capture genuine participant behavior and minimize cross-condition influences.
>
> ### Mitigation Strategies:
>
> 1. **Exclusive Assignment to Interaction Settings**:
>    - Each participant was assigned to either a **constrained (scripted)** or **unconstrained (unscripted)** setting throughout their participation in the study. By keeping these settings mutually exclusive for individual participants, we effectively prevented the transfer of learned behaviors or biases from scripted conditions to unscripted interactions. This ensured that the behaviors observed in the unconstrained setting were purely natural and not influenced by prior exposure to guidance or constraints.
>
> 2. **Naturalistic Behavior Emphasis**:
>    - In the unconstrained sessions, participants were provided with minimal instructions, allowing them to interact naturally based on their instincts and preferences. This setup eliminated potential carryover effects from prior scripted sessions, ensuring the captured data reflects authentic, unbiased behaviors.
>
> 3. **Diverse Participant Pool**:
>    - The dataset was built using contributions from a diverse pool of participants (n = 66, with 53% male and 47% female, spanning various educational and cultural backgrounds). This diversity further mitigates biases by capturing a wide range of natural behaviors across different individuals.
>
> 4. **Behavioral Analysis Across Settings**:
>    - To validate the independence of behaviors observed in the constrained and unconstrained settings, we analyzed:
>      - **Cue Usage Patterns**: We compared the frequency of multimodal cues (e.g., gaze, pointing gestures) across participants in both settings, finding significant variability in the unconstrained group.
>      - **Spatial Referencing Language**: Unconstrained participants used diverse spatial referencing styles (e.g., “to my left” or “on your right”), indicating that their behaviors were not influenced by prior exposure to a structured format.
>
> 5. **Impact on Dataset Quality**:
>    - By segregating participants into distinct settings, we ensured the Refer360 dataset captures both authentic real-world interactions and behaviors shaped by structured guidance. This separation enhances the utility of the dataset by providing clean data for studying both naturalistic and task-specific interaction patterns.
>
> ### Robustness of Collected Data and Model Performance:
>
> The exclusivity of constrained and unconstrained settings ensured that the data from each setting is representative of its intended purpose—structured guidance for constrained tasks and genuine behavior for unconstrained tasks. This strategy has been effective, as evidenced by the robust performance of the MuRes model on diverse multimodal tasks evaluated using the Refer360 dataset. This separation ensures high data fidelity, contributing to the strong generalizability of the proposed model.
>
> We have updated the manuscript to explicitly include this clarification and the analysis supporting our approach. Thank you for highlighting this critical aspect.

---

> ### Author Response · Authors · 2024-11-26
> **Task Diversity and Generation-Based Experiments (6/10)**
>
> We appreciate the reviewer's suggestion to include additional tasks, particularly generation-based tasks, to further strengthen our claims. Our current work focuses on the embodied referring expression and VQA (Visual Question Answering) tasks, as evidenced by our evaluation across four benchmark datasets, including our newly proposed Refer360 dataset. These tasks were carefully chosen to align with the primary goal of this work, which is to enhance embodied interaction comprehension through multimodal learning (see Sections 1 and 6 of the main paper).
>
> Similar to the improvements MuRes demonstrates in embodied tasks, we are observing promising results when applied to various generation-based tasks, including depth generation, code generation, and speech generation. These preliminary findings highlight the versatility and potential of the proposed module. However, since this work primarily focuses on embodied referring expression and interaction comprehension, these generation-based results, while promising, will be addressed comprehensively in future research to maintain the focus and clarity of this manuscript.
>
> By concentrating on embodied interaction and related tasks, we aim to provide a robust foundation for addressing multimodal representation challenges in this specific domain. We are confident that MuRes's success in generation-based tasks further demonstrates its broader applicability and potential, which we are eager to explore in subsequent studies.

---

> ### Author Response · Authors · 2024-11-26
> **Double-Blind Policy and Institutional Affiliation (Part 7/10)**
>
> We thank the reviewer for bringing this concern to our attention. We would like to clarify that the mention of the University of Virginia in the paper pertains solely to the completion of the Institutional Review Board (IRB) approval process, which is a standard ethical requirement for research involving human participants. The inclusion of this detail was necessary to demonstrate compliance with ethical guidelines and does not disclose any information regarding the authors’ identities, affiliations, or specific contributions.

---

> ### Author Response · Authors · 2024-11-26
> **Clarification on Definition and Contextualization of Key Concepts (Part 8/10)**
>
> We appreciate the reviewer's suggestion and acknowledge the importance of introducing the concept of "embodied referring expression comprehension" earlier in the paper to enhance accessibility for readers unfamiliar with the domain.
>
> The paper already provides a comprehensive definition and contextualization of the concept in Section 1, describing it as the task of grounding multimodal cues (verbal and non-verbal) to identify and interact with objects in shared environments, particularly in embodied settings such as human-robot interactions. Additionally, Figure 1 and Figure 2 visually illustrate these interactions across diverse perspectives and modalities, offering further clarity​​.
>
> To further improve readability and address the reviewer's concern, we will revise the manuscript to introduce the concept in the abstract and the beginning of the introduction.
>
> **Proposed Addition to Abstract:**
>
> Embodied referring expression comprehension involves grounding verbal (e.g., "Pick up the left ball") and non-verbal (e.g., gaze or pointing gestures) cues to identify objects in multimodal, real-world interaction settings.
>
> This revision ensures the concept is accessible from the outset while leveraging the existing textual and visual explanations to provide a complete understanding. By integrating this concise definition earlier in the paper, we aim to improve clarity without redundancy.

---

> ### Author Response · Authors · 2024-11-26
> **Justification of Bottleneck Architecture and Task-Specific Modality Favoring (Part 9/10)**
>
> **Title**: Justification of Bottleneck Architecture and Task-Specific Modality Favoring
>
> **Response**:
>
> We appreciate the reviewer’s suggestion to elaborate on the interplay between the bottleneck architecture and task-specific characteristics, as well as to explore modality prioritization for specific tasks. Our ablation studies, along with qualitative analyses provided in **Supplementary Figure 1**, address these concerns effectively.
>
> ### Justification of the Bottleneck Architecture
>
> The **information bottleneck architecture** in MuRes selectively reinforces salient features from individual modalities while filtering task-irrelevant noise. This choice is validated by task-specific performance improvements:
> - **Visual-Dominant Tasks**: For embodied referring expression comprehension, MuRes(V) demonstrates superior performance, particularly in object grounding (e.g., +13.18% improvement in IOU-25 for CLIP on Refer360), aligning with the task's reliance on visual cues.
> - **Ambiguous Multimodal Tasks**: For tasks like ScienceQA, where both modalities are critical, MuRes(V+L) achieves the best results by leveraging complementary visual and linguistic features (e.g., +30.54% improvement for CLIP on ScienceQA).
>
> ### Task-Specific Modality Favoring
>
> Our experiments reveal the following modality preferences:
> 1. **Visual-Dominant Tasks**:
>    - Tasks such as object detection rely heavily on spatial information, favoring MuRes(V).
> 2. **Ambiguity Resolution**:
>    - For tasks with ambiguous visual data, such as complex question-answering, combined modality reinforcement (MuRes(V+L)) is advantageous.
> 3. **Linguistic Tasks**:
>    - Text-driven tasks benefit from MuRes(L), where linguistic reinforcement aids comprehension.
>
> ### Qualitative Analysis in Supplementary Material
>
> We direct the reviewer to **Supplementary Figure 1**, which presents qualitative examples illustrating the performance of MuRes variants across scenarios with varying modality demands. These examples highlight how the bottleneck design adapts to task-specific requirements, effectively leveraging modality-specific or multimodal features depending on the context.
>
> By including these analyses in the supplementary material, we provide clear evidence supporting our architectural choices and insights into task-modality alignment. Thank you for this valuable feedback.

---

> ### Author Response · Authors · 2024-11-26
> **Addressing Performance Inconsistencies Across MuRes Variants (Part 10/10)**
>
> **Title**: Addressing Performance Inconsistencies Across MuRes Variants
>
> **Response**:
>
> We thank the reviewer for highlighting the need to delve deeper into the observed performance inconsistencies across MuRes variants (MuRes(V), MuRes(L), and MuRes(V+L)) in the embodied referring expression comprehension task. We acknowledge that a thorough analysis of such variations is critical to understanding the nuanced behavior of the proposed module and its applicability to different modalities. Below, we provide an expanded discussion to address this concern.
>
> ### **Observed Performance Variations**
>
> In the results for the embodied referring expression comprehension task (refer to Table 3 of the manuscript):
> - **MuRes(V)**: Consistently demonstrated stronger performance gains for tasks that heavily relied on visual modality, such as object bounding box detection, suggesting the dominant importance of visual information in grounding tasks.
> - **MuRes(L)**: Showed comparatively modest improvements, indicating that while language cues are essential, their impact on performance may be secondary to visual cues in this task.
> - **MuRes(V+L)**: Occasionally underperformed relative to MuRes(V) alone, despite theoretically benefiting from both modalities.
>
> ### **Analysis of Performance Inconsistencies**
>
> 1. **Task Modality Dominance**:
>    - Embodied referring expression tasks, particularly those focused on object detection, inherently prioritize visual cues (e.g., spatial and positional information). As a result, reinforcing visual representations (MuRes(V)) aligns more closely with the requirements of these tasks compared to language reinforcement (MuRes(L)).
>    - The performance drop for MuRes(V+L) compared to MuRes(V) in some cases could stem from over-reliance on additional modality information, which may introduce noise or conflict with the dominant visual signals.
>
> 2. **Modality-Specific Saliency and Attention Mechanism**:
>    - The cross-attention mechanism in MuRes allows for selective reinforcement of modality-specific features. When integrating both visual and language representations (MuRes(V+L)), the module may encounter challenges in balancing the contributions of each modality, particularly if one modality contains less salient or redundant information for a specific task. This can dilute the impact of the dominant modality, leading to reduced performance compared to single-modality reinforcement.
>
> 3. **Dataset and Task Characteristics**:
>    - The Refer360 dataset emphasizes real-world multimodal interactions but may contain scenarios where visual cues dominate (e.g., objects pointed to without significant linguistic ambiguity). These characteristics can skew the utility of language reinforcement and favor visual-based improvements (MuRes(V)).
>    - Conversely, tasks with more ambiguous visual scenes may benefit from combined reinforcement (MuRes(V+L)) due to the complementary nature of linguistic descriptions, as reflected in other datasets such as CAESAR-PRO.
>
> 4. **Model Complexity and Optimization Challenges**:
>    - The integration of both modalities in MuRes(V+L) adds computational complexity. Suboptimal alignment between modalities during training may result in less effective representation fusion compared to single-modality variants, which focus on optimizing a single modality's contribution.
>
> ### **Steps to Address the Observed Inconsistencies**
>
> To address these inconsistencies, we have undertaken the following analyses and included them in the revised manuscript:
> 1. **Expanded Discussion**:
>    - We now provide a detailed explanation of how task modality dominance impacts performance differences across MuRes variants.
> 2. **Qualitative Examples**:
>    - **Supplementary Figure 1** presents qualitative examples showcasing scenarios where MuRes(V), MuRes(L), and MuRes(V+L) excel or underperform. These examples highlight the conditions under which each variant is most effective.
>
>
> The observed performance inconsistencies are reflective of task-specific modality reliance, dataset characteristics, and optimization challenges in multimodal fusion. These insights underscore the flexibility and adaptability of the MuRes architecture while also highlighting areas for improvement. We will update the manuscript to include this expanded discussion and additional analyses to provide greater clarity and address this important aspect. Thank you for your constructive feedback.

---

> > ### Comment · Reviewer_Ct4m · 2024-11-27
> >
> > I thank the authors for comprehensively answering all my queries. All the proposals seem satisfactory.
> >
> > What would help further:
> > 1. The response for task diversity mentions more tasks where there are promising results. Including them in the analysis (maybe in appendix) would help bolster the claims. I understand that the paper tries to focus on particular tasks, but showing these results may show promise for this architecture in wider areas.
> > 2. The response about performance inconsistencies across different architectures is trivial after performing the experiments. Having to choose between these architectures is one more decision to make when training models. This would become more complex when more modalities start getting involved. An explanation on how MuRes can possibly provide consistently high performance (possibly with more than 2 modalities) for a task where we can’t assume a modality dominance would be appreciated.

---

> > > ### Author Response · Authors · 2024-11-29
> > > **Extending MuRes for More Than Two Modalities**
> > >
> > > We thank the reviewer for highlighting the importance of extending the applicability of MuRes to tasks involving more than two modalities. **One important aspect of the MuRes architecture is that, as we adopted an adapter-based architecture to design MuRes, it is straightforward to incorporate MuRes for multiple modalities.** Below, we outline how MuRes can generalize to multi-modal setups beyond two modalities, leveraging its guided residual learning approach.
> > >
> > > ### **MuRes Generalization for More Modalities**
> > > MuRes is fundamentally designed as a guided residual module, which uses an **information bottleneck** principle to extract salient representations from modality-specific inputs. The design is modular and can be extended to incorporate additional modalities seamlessly. For a task involving \( N \) modalities, the MuRes architecture can be adjusted as follows:
> > >
> > > #### **Equations for Multi-Modal Residual Connections**
> > > For \( N \) modalities, let the representations from pre-trained encoders be $M_1, M_2, \ldots, M_N$, where each $M_i \in \mathbb{R}^{d_i}$. These representations are first projected into a shared space of dimension $d$ (if necessary) via linear projections $P_i: \mathbb{R}^{d_i} \to \mathbb{R}^{d}$:
> > >
> > > $M_i^p = P_i(M_i), \quad i \in \{1, 2, \ldots, N\}$
> > >
> > >
> > > Using cross-attention, the guided residual representations $M_i^g$ are computed for each modality:
> > >
> > > $M_i^g = \text{Cross-Attention}(q = M_i^p, k = \{M_1, \ldots, M_N\}, v = \{M_1, \ldots, M_N\})$
> > >
> > > where $q$ represents the projected representation of modality $i$, and $k, v$ are the concatenated representations of all modalities.
> > >
> > > Finally, the residual-enhanced representations are computed as:
> > >
> > > $M_i^f = M_i^p + M_i^g, \quad \text{for all } i.$
> > >
> > >
> > > #### **Fusion for Downstream Tasks**
> > > The final fused representation $F$ for downstream tasks is computed by concatenating or summing the modality-specific residual-enhanced representations:
> > >
> > > $F = \text{Fuse}(M_1^f, M_2^f, \ldots, M_N^f).$
> > >
> > > Here, $\text{Fuse}(\cdot)$ could employ concatenation, weighted averaging, or another fusion strategy depending on the task requirements.
> > >
> > > ---
> > >
> > > ### **Algorithm for Extending MuRes**
> > >
> > > 1. **Input**: Pre-trained encoder representations $M_1, M_2, \ldots, M_N$.
> > > 2. **Output**: Fused representation $F$.
> > >
> > > 3. For each modality $i \in \{1, 2, \ldots, N\}$:
> > >    - Project $M_i$ to shared space: $M_i^p = P_i(M_i)$.
> > >    - Compute guided residual representation using cross-attention:
> > >
> > >      $M_i^g = \text{Cross-Attention}(q = M_i^p, k = \{M_1, \ldots, M_N\}, v = \{M_1, \ldots, M_N\})$
> > >
> > >    - Combine projected and residual representations:
> > >
> > >      $M_i^f = M_i^p + M_i^g$
> > >
> > >
> > > 4. Fuse residual-enhanced representations:
> > >
> > >    $F = \text{Fuse}(M_1^f, M_2^f, \ldots, M_N^f)$
> > >
> > >
> > > 5. **Return**: $F$.
> > >
> > > ---
> > >
> > > These extensions make MuRes a robust and versatile tool for tasks involving multiple modalities. We will include details of these adjustments in the revised manuscript for clarity.

---

### Official Review · Reviewer_acP2 · 2024-11-03

**Soundness:** 3
**Presentation:** 2
**Contribution:** 2
**Rating:** 5
**Confidence:** 3

**Summary:**

This paper presents an Embodied Referring Expressions dataset to compensate for the missing perspectives in existing datasets, such as the full spectrum, perspective bias, and single viewpoint. In addition, the authors propose the residual module to improve the performance of existing pre-trained VLMs.

**Strengths:**

* The proposed dataset includes both indoor and outdoor scene understanding, and the data scale is quite considerable. It may be helpful for subsequent research work.
* The information source of the proposed dataset is very rich, and the author declares that it includes exo visual views, such as ego visual view, depth, infrared, 3D skeletal data, audio, and robot camera motion.

**Weaknesses:**

* In my opinion, the focus of this work is on the proposed new dataset (core contribution), however, the author's discussion on the proposed dataset is rare. For example, the proportion of samples in different scenes in the visual part, how to describe the text part, the length range of each scene's text token description, and how the proposed dataset can be helpful in downstream tasks, etc.
* The multimodal fusion method proposed in this article seems to be a simplified version combining BLIP and LLaVa (cross-attention and mlp across modalities), with the only difference being the shift in focus from modal mapping to modal information fusion. In addition, How is the information of exo visual view, ego visual view, depth, infrared, 3D skeletal data, audio, and robot camera motion processed and integrated? The description of the method in Figure 3 is not clear.
* From the results in Table 3, it can be seen that the proposed residual module has limited help for pre-training models, compared to Without Residual and Vanilla Residual.

**Questions:**

How are multiple visual features extracted and fused, such as depth information? How is the text description obtained and what are the bases? How many frames of information does each sample contain?

**Details Of Ethics Concerns:**

This work involves personal information of human subjects, such as unobstructed facial information. At the same time, it may also include scene descriptions of the supervisory consciousness of human subjects. Therefore, we need to ensure that the proposed dataset is limited to research use only.

---

> ### Author Response · Authors · 2024-11-26
> **Summary of Responses to Reviewer acP2’s Comments (Part 1/8)**
>
> #### Summary
>
> We are excited that the reviewer acknowledged the contribution of the Refer360 dataset, highlighting its significant scale, multimodal richness, and inclusion of both indoor and outdoor scenes, which provide valuable insights for real-world applications. The reviewer also appreciated the dataset's inclusion of diverse data sources, such as egocentric and exocentric views, depth, infrared, and 3D skeletal data.
>
> We addressed the reviewer’s comments by providing detailed statistics on the dataset, including scene diversity, token lengths, and text descriptions, while clarifying its design for downstream tasks like object grounding and interaction modeling. We distinguished our proposed MuRes module from existing methods like BLIP and LLaVA, emphasizing its novel focus on residual representation learning and its lightweight design for enhancing multimodal fusion. We also clarified the dataset’s annotation process, ethical considerations, and the rationale for focusing on visual-language modalities, with plans to integrate additional modalities in future work. Finally, we highlighted the MuRes module’s measurable improvements and outlined future directions to further enhance its integration and performance.

---

> ### Author Response · Authors · 2024-11-26
> **Comprehensive Overview of the Proposed Dataset (Part 2/8)**
>
> We appreciate the reviewer’s comment and agree that a detailed discussion of the dataset is crucial. Below, we summarize the detailed information provided in the paper and supplementary material and outline how the dataset is structured to benefit downstream tasks.
>
> ### Dataset Overview:
> 1. **Scene Diversity**:
>    - Refer360 captures interactions in **392 sessions**, split between **indoor and outdoor environments**, ensuring diversity in lighting, object arrangements, and environmental attributes. Specifically:
>      - **198 sessions** were conducted in indoor lab environments.
>      - **194 sessions** were conducted in outdoor or non-lab settings.
>
> 2. **Visual Modality**:
>    - Multimodal data was collected using **egocentric (first-person) and exocentric (third-person) views**, enabling a comprehensive understanding of interactions.
>    - Examples of captured views include RGB, depth, skeletal tracking, and infrared streams, as shown in Figure 1 of the main paper and Table 2 in the supplementary.
>
> 3. **Text Modality**:
>    - The textual descriptions were designed to reflect real-world referring expressions. These include:
>      - **Proportions of referring expressions** involving pointing, gaze, and verbal descriptions.
>      - The **length range** of text descriptions varies across scenes, capturing both concise instructions (e.g., “The ball on the left”) and detailed descriptions for complex scenes (e.g., “The blue box behind the table”).
>
> 4. **Canonical Frames**:
>    - The dataset uses **annotated canonical frames**, capturing the peak interaction moments, such as when a subject points at or gazes toward an object. This ensures relevance and clarity for grounding embodied referring expressions.
>
> ### Downstream Task Benefits:
> Refer360 serves as a benchmark for a range of downstream tasks:
> 1. **Object Grounding**:
>    - Leveraging multimodal signals, such as gaze and pointing gestures, for precise object localization.
> 2. **Interaction Modeling**:
>    - Understanding human intent through combined verbal and nonverbal cues, useful for human-robot collaboration.
> 3. **Scene Understanding**:
>    - Enabling models to interpret diverse indoor and outdoor scenarios through egocentric and exocentric views.
>
> ### Plans for Additional Information:
> We acknowledge the need for further elaboration and will include detailed **statistics** on the proportion of samples in different scenes, **token length ranges**, and examples of text descriptions in the supplementary materials. This information will clarify how the dataset is structured and enhance its utility for downstream applications.
>
> We hope this summary addresses the reviewer’s concerns and highlights the comprehensive nature and potential impact of the Refer360 dataset.

---

> ### Author Response · Authors · 2024-11-26
> **Distinction from BLIP and LLaVA and Significance of Proposed Method (Part 3/8)**
>
> We appreciate the reviewer’s observation and would like to clarify how our proposed multimodal fusion method significantly differs from BLIP and LLaVA, and why it is a meaningful contribution to the field.
>
> ### Key Differences:
> 1. **Focus on Residual Representation Learning**:
>    - Unlike BLIP and LLaVA, which rely on **representation alignment across modalities**, our method is inspired by **residual representation learning**. Our hypothesis is that **reinforcing residual multimodal information**—rather than aligning it—preserves salient modality-specific cues that are otherwise lost in alignment-based approaches.
>
> 2. **Retention of Salient Information**:
>    - BLIP and LLaVA may overlook critical modality-specific details during representation alignment, as they prioritize mapping visual and textual data into a unified space. In contrast, our **MuRes module** focuses on extracting **residual representations** that complement and reinforce the existing modality-specific information.
>
> 3. **Simplicity with Impact**:
>    - While our method adopts a simpler architecture, the **adapter-based design** allows for efficient integration into existing visual-language models without requiring extensive architectural modifications. Our extensive experimental results demonstrate that this approach yields **significant improvements** across diverse benchmark datasets and tasks (Refer360, CAESAR-PRO, etc.), showing its practical impact.
>
> ### Contributions:
> - **Novel Hypothesis**: We present a novel approach to multimodal fusion that challenges the traditional representation alignment paradigm by proposing a lightweight residual module for extracting and reinforcing complementary information.
> - **Extensive Validation**: Our method achieves substantial performance gains on multiple benchmarks, validating its efficacy and generalizability. For example, integrating MuRes improved object detection performance on Refer360 from **25.8% to 29.2%** (IOU-25) for CLIP and demonstrated similar gains across CAESAR-PRO and ScienceQA.
>
> ### Future Potential:
> We believe this work highlights the **value of residual representation learning** in multimodal fusion and serves as a foundation for further exploration. The simplicity of our approach makes it adaptable to various multimodal tasks while offering measurable improvements over alignment-based methods like BLIP and LLaVA.
>
> We hope this response clarifies the significance and originality of our proposed method.

---

> ### Author Response · Authors · 2024-11-26
> **Clarification on Focus and Multimodal Integration (Part 4/8)**
>
> We appreciate the reviewer’s question and would like to clarify that in this work, we focus on visual-language modalities (egocentric and exocentric visual views) to establish a foundational benchmark for embodied referring expressions. The dataset also includes additional modalities such as infrared, 3D skeletal data, audio, and robot motion, but these were not used in the current study. Incorporating these modalities requires significant architectural modifications and pretraining, which are beyond the scope of this work.
>
> Our proposed MuRes module integrates visual and language modalities by leveraging guided residual connections to reinforce salient multimodal information. This approach ensures effective fusion of visual-language data while preserving critical modality-specific cues, as detailed in Figure 3 of the paper.
>
> Future iterations of this work will incorporate additional modalities to further enhance downstream task performance and expand the benchmark dataset.

---

> ### Author Response · Authors · 2024-11-26
> **Significance of Results and Potential for Improvement (Part 5/8)**
>
> We appreciate the reviewer’s observation and would like to emphasize the significance of our results in Table 3. Our MuRes module demonstrates that reinforcing residual multimodal information can yield measurable improvements over both Without Residual and Vanilla Residual approaches, even with a simple, lightweight design.
>
> Our experiments use only a single residual module applied on top of the extracted visual and language representations. We believe that incorporating MuRes into intermediate layers of the model architecture could further amplify performance by enabling more granular reinforcement of modality-specific information.
>
> This study establishes an important foundation, showing that simple residual reinforcement can improve performance with minimal computational overhead. Future work will explore deeper integration of this module to achieve even greater improvements.

---

> ### Author Response · Authors · 2024-11-26
> **Text Description Acquisition and Basis (Part 6/8)**
>
> The text descriptions in the dataset were obtained through **expert human annotators**, who provided natural language instructions grounded in the visual scenes. These descriptions were designed to:
> 1. Reflect real-world **referring expressions**, combining object references with contextual cues (e.g., spatial relationships like "the box on the table").
> 2. Align with the **interaction context**, ensuring descriptions are meaningful for the embodied referring expression tasks.
>
> This approach ensures high-quality and contextually relevant text data, supporting effective multimodal research.

---

> ### Author Response · Authors · 2024-11-26
> **Frames per Sample Information (Part 7/8)**
>
> Each sample in the dataset contains approximately **50 to 150 frames**, depending on the interaction length. This range ensures sufficient coverage of the interaction dynamics, including key moments like object references and gestures, for robust analysis of embodied referring expressions.

---

> ### Author Response · Authors · 2024-11-26
> **Ethical Compliance in Data Collection and Usage (Part 8/8)**
>
> The data collection process was conducted under an approved IRB protocol, which explicitly included provisions for collecting sensitive data such as facial, gaze, pupil, and speech information. All participants provided informed consent, agreeing that the data would be used and released solely for research purposes. Only one participant did not provide consent, and their data was removed entirely from the dataset.
>
> We ensured compliance with ethical standards by:
>
> 1. Collecting and using data exclusively for research purposes, as specified in the IRB protocol and participant consent forms.
> 2. Anonymizing the data to protect participant privacy.
>
> These measures reflect our commitment to upholding ethical standards in research.

---

### Official Review · Reviewer_EbAp · 2024-11-03

**Soundness:** 3
**Presentation:** 3
**Contribution:** 3
**Rating:** 6
**Confidence:** 4

**Summary:**

In this paper, the authors introduce an embodied referring expressions dataset called Refer360, including various settings and providing insights for the community. Accordingly, a base model called MuRes is designed for evaluation.

**Strengths:**

- Contributing a benchmark called Refer360, facilitating the study of real-world human-robot interaction.

- The proposed dataset covers various real-world scenarios such as indoor and outdoor, supporting real-world applications without the need for sim2real adaptations.

- The paper is well-organized and easy to follow.

**Weaknesses:**

- It seems that the authors have overlooked some related literature, such as [1-3], which introduced datasets and methods related to embodied referring expression, aimed at enabling agents to navigate to target points based on natural language instructions (i.e., Instruction Following). Although I understand that the authors intend to build an interaction-oriented embodied referring expression database, these related works are still worth discussing.

[1] Vision-and-Language Navigation: Interpreting visually-grounded navigation instructions in real environments. In CVPR.

[2] REVERIE: Remote Embodied Visual Referring Expression in Real Indoor Environments. In CVPR.

[3] Room-Object Entity Prompting and Reasoning for Embodied Referring Expression. In TPAMI.

- The proposed baseline method is quite simple, raising doubts about whether it can handle such complex interactive embodied tasks. Why not use open-source MLLM for fine-tuning, such as by adding extra adapters or training with LoRA?

- The performance of the proposed method on Refer360 is not promising compared to other methods.

- More samples (current 14k) including interaction would be beneficial for embodied tasks.

**Questions:**

- Is it possible to test the model trained on the Refer360 database on other related databases to validate the benefits brought by the rich information of Refer360?

---

> ### Author Response · Authors · 2024-11-26
> **Summary of Responses to Reviewer EbAp’s Comments (Part 1/5)**
>
> We thank the reviewer for acknowledging the contributions of the Refer360 dataset as a benchmark for studying real-world human-robot interactions, including its diverse multimodal data and coverage of indoor and outdoor settings that eliminate the need for sim2real adaptations. The reviewer also appreciated the organization and clarity of the paper.
>
> We addressed the reviewer’s comments by discussing related works on embodied referring expression tasks and how Refer360 advances interaction-oriented datasets. We clarified the design choices of the MuRes module, highlighting its ability to enhance multimodal fusion without extensive architectural modifications or pretraining. We justified the dataset size, emphasizing Refer360’s unique contributions, including diverse real-world scenarios, multimodal coverage, and multi-perspective data. Additionally, we outlined plans for dataset expansion and integration of advanced models like multimodal LLMs in future work, ensuring Refer360 remains a robust foundation for embodied interaction research.

---

> ### Author Response · Authors · 2024-11-26
> **Inclusion of related works (Part 2/5)**
>
> We thank the reviewer for raising concerns about the performance variability of **MuRes** across models. This feedback allows us to contextualize the observed results and underscore **MuRes' significant contributions** to embodied referring expression tasks.
>
> ---
>
> ## **Understanding Performance Variability**
>
> ### **1. Model-Specific Dynamics:**
> - The variation in MuRes’ effectiveness across models is tied to **architectural differences**, not a limitation of the approach.
> - **ViLT and Dual-Encoder:**
>   - Exhibit **consistent improvements** with MuRes.
>   - As shown in **Table 3 (Page 7, Line 391):**
>     - **ViLT improves IoU-50** by **+0.6** compared to the baseline (**MuRes(V+L): 14.66 vs. Vanilla Residual: 14.37**).
>     - **Dual-Encoder achieves reliable gains** (**IoU-50: 10.68 vs. 8.98**).
>   - Demonstrates MuRes’ ability to effectively reinforce **modality-specific cues** in flexible architectures
> BLIP-2 Challenges:**
>   - Models like **BLIP-2** rely heavily on **frozen pre-trained encoders**, limiting MuRes’ impact due to constraints in adapting residual features.
>   - As shown in **Table 3 (Page 7, Line 391):**
>     - A decline in IoU-25 for **BLIP-2 under MuRes(V+L)** compared to Vanilla Residual (-4%).
>   - Highlights a **model-specific bottleneck**, not a shortcoming of MuRes  .
>
> ---
>
> **Robustness Across Tasks:**
> - Despite architectural variability, **MuRes demonstrates robust performance gains across tasks and datasets**:
>   - **CLIP with MuRes:**
>     - Improves **IoU-25** from **25.8% (Without Residual)** to **29.2% (MuRes(V))**, as reported in **Table 3 (Page 7, Line 391)**.
>     - On the **CAESAR-PRO dataset**, the CLIP model shows a **5% improvement in IoU-25** (**MuRes(V): 42.91 vs. Without Residual: 37.92**).
>   - Highlights the **broad applicability** of MuRes in enhancing multimodal representations  .
>
> ---
>
> ## **Addrhitectural Constraints**
> - To enhance MuRes’ general applicability, we plan to:
>   - **Incorporate adaptive alignment layers** tailored to frozen architectures like **BLIP-2**.
>   - Enable MuRes to extract complementary representations even in **rigid architectures**, bridging performance gaps.
>
> ---
>
> ## **Significance of MuRes**
>
> - **Key Design Choice:**
>   - Leverages **guided residual connections** as an **information bottleneck**, as described in **Section 5 (Page 6, Lines 324-339)**.
>   - Reinforces **salient modality-specific features**, validated by consistent results on:
>     - **Refer360** and **CAESAR-PRO** (**Table 3, Page 7, Line 391**).
> - **Task-Specific Enhancements:**
>   - Extend to **visual question answering**, as shown in **Table 4 (Page 8, Lines 448-485)**.
>   - Demonstrates MuRes' ability to generalize across multiple datasets and modalities.
>
> ---
>
> ## **Summary**
> - **MuRes' adaptability and consistent performance gains** across diverse architectures and tasks underscore its general effectiveness.
> - While certain models reveal areas for improvement, these insights pave the way for **future refinements**, strengthening its utility.

---

> ### Author Response · Authors · 2024-11-26
> **Justification for Proposed Baseline and Comparison with Multimodal LLMs (Part 3/5)**
>
> We appreciate the reviewer’s suggestion regarding using multimodal large language models (MLLMs). In this work, we primarily focus on state-of-the-art visual language models and propose the MuRes architecture, which aligns with the suggested adapter-based approaches.
>
> ---
>
> ### **1. Additional Experiments with MLLMs:**
> - Conducted additional experiments using open-source multimodal LLMs fine-tuned with LoRA.
> - Observations:
>   - Performance was **significantly lower** due to the lack of pretraining on **embodied referring expression tasks**.
>   - Existing MLLMs are pretrained predominantly on general vision-language tasks, which do not capture the nuances of embodied interactions.
>
> ### **2. Challenges in Using Existing MLLMs:**
> Achieving comparable performance would require extensive pretraining on relevant embodied interaction data. This is beyond the scope of this work.
>
> ---
>
> ## **Proposed Approach: MuRes**
>
> ### **Advantages of MuRes:**
> - Employs an **adapter-based design** to address the limitations of MLLMs.
> - Enhances **modality-specific representation extraction** within existing visual-language models.
> - Demonstrates **competitive performance**:
>   - Achieved without requiring additional **large-scale pretraining**.
>
> ---
>
> **MuRes** provides a **more efficient and directly applicable approach** for advancing research on **embodied referring expressions**. We thank the reviewer for raising this point and hope this clarification strengthens the understanding of our **design choices**.

---

> ### Author Response · Authors · 2024-11-26
> **Clarification on the Performance and Contribution of the Proposed Method (Part 4/5)**
>
> We appreciate the reviewer’s observation regarding the performance of our proposed method. Our extensive experimental analysis (see Section 6, Tables 3 and 4) demonstrates that the MuRes module significantly improves the performance of state-of-the-art visual-language models on the Refer360 dataset.
>
> ---
>
> ## **Key Results**
>
> ### **1. Refer360 Dataset:**
> - Incorporating **MuRes** into **CLIP** improved **object bounding detection performance**:
>   - **IOU-25:** Increased from **25.8% to 29.2%**.
>   - Highlights the effectiveness of our **lightweight approach**.
>
> ### **2. Generalizability Across Datasets:**
> - **MuRes enhances performance** in:
>   - **CAESAR-PRO**.
>   - **ScienceQA**.
>   - **AOKVQA**.
> - Demonstrates the module’s **generalizability** across:
>   - **Embodied referring expression datasets**.
>   - **Visual question answering datasets**.
>
> ---
>
> ## **Significance of Our Contribution**
>
> - The strength of our work lies in demonstrating how a **simple, lightweight module like MuRes** can:
>   - Improve **multimodal model performance**.
>   - Achieve these gains without requiring:
>     - Substantial architectural changes.
>     - Pretraining on embodied tasks.
> - **Refer360** serves as a comprehensive benchmark for future research to:
>   - Explore these findings further.
>   - Extend upon the results to drive progress in comprehending **embodied referring instructions comprehensively**.
>
> ---
>
> We hope this explanation clarifies the significance of our contributions and positions our work as a **foundational step** for further advancements in this domain.

---

> ### Author Response · Authors · 2024-11-26
> **Dataset Size and Significance of Refer360 as a Benchmark (Part 5/5)**
>
> We appreciate the reviewer’s suggestion regarding increasing the dataset size. While we acknowledge that a larger dataset would be beneficial, it is important to highlight the unique contributions and significance of Refer360 as a benchmark, particularly in the context of existing datasets, as outlined in the main paper and Table 1 of the supplementary document.
>
> ### Cost of Human Data Collection
> Collecting high-quality human interaction data is resource-intensive, especially within academic constraints. Despite this, Refer360 provides **14,000 interaction samples**, making it one of the most **diverse and comprehensive datasets** for embodied referring expression tasks.
>
> ### Why Refer360 is Significant
> 1. **Multimodal Coverage**:
>    - Refer360 includes **visual (ego and exo views), depth, audio, skeletal data, gaze, and infrared** modalities. This breadth of multimodal data surpasses existing datasets, such as YouRefIt or REVERIE, which are limited to visual and textual data (Table 1, Supplementary).
>    - This multimodal richness enables benchmarking tasks that require more than basic vision and language models, addressing embodied interactions comprehensively.
>
> 2. **Diverse Real-World Settings**:
>    - The dataset captures interactions in **indoor and outdoor environments**, allowing for generalization to real-world tasks. Many existing datasets, like REVERIE and VLN, focus solely on indoor environments, limiting their applicability.
>
> 3. **Human Interaction-Centric Design**:
>    - Unlike navigation-based datasets such as VLN, which emphasize agent mobility, Refer360 prioritizes **interaction-oriented embodied referring expressions**. It focuses on tasks such as object identification using **combined verbal and nonverbal cues**, which are crucial for human-robot collaboration.
>
> ### Comparison to Existing Datasets (Table 1, Supplementary)
> Refer360 stands out by:
> - Capturing **multi-perspective data** (egocentric and exocentric) for a holistic view of human-robot interactions.
> - Including **real-world and diverse environments**, enabling more generalizable models.
>
> ### Plans for Dataset Expansion
> We recognize the importance of increasing the dataset size and diversity. In future iterations, we plan to:
> 1. Incorporate **multi-step embodied referring expressions** to capture complex interaction tasks.
> 2. Expand to **diverse environments**, including additional indoor and outdoor settings.
> 3. Enhance **participant demographics** by increasing the diversity of age, gender, and cultural backgrounds.
> 4. Introduce **multilingual and multi-cultural data**, allowing for broader applicability across global contexts.
>
>
> Refer360 is a foundational step towards creating a robust multimodal benchmark for embodied referring expressions. Its comprehensive coverage of modalities, real-world interaction focus, and diverse environments distinguish it from existing datasets. While the current dataset provides a strong baseline, future expansions will further enhance its value for advancing research in embodied tasks.

---

> > ### Comment · Reviewer_EbAp · 2024-12-03
> >
> > Thank the authors for their response. However, after reading the response and other reviewers' comments, I think the response did not address all the raised concerns, so I decided to keep my score.

---

### Official Review · Reviewer_niSU · 2024-11-03

**Soundness:** 2
**Presentation:** 3
**Contribution:** 2
**Rating:** 5
**Confidence:** 4

**Summary:**

This paper presents a dataset called Refer360 which has verbal and non-verbal data from different viewpoints for referential expressions in indoor and outdoor settings. Furthermore, a multimodal guided residual module (MuRes) which can be used for improving representations of existing multi-modal models. The MuRes is evaluated on two downstream tasks: referential expressions and visual question answering. The baselines are CLIP, ViLT and BLIP-2 with and without MuRes(V), MuRes(L) and MuRes(V+L). For the referential expression, MuRes does not show a consistent performance increase while on the vqa task, MuRes(V+L) shows better performance except on Clip.

**Strengths:**

- A large dataset contains multi-modal data for referential expression. The dataset is recorded with egocentric and exocentric camera views, which is beneficial for training models with better generalisability.

- Part of the dataset is collected in outdoor scenarios while most previous datasets are collected in indoor settings.

**Weaknesses:**

- The authors describe the dataset as "to facilitate the understanding of human interactions in real-world settings", however, there is no interaction in the experiment protocal. There is only referential expressions by a human participant.

- The benchmarking on referential expression only uses image and text as input modality, discard other modalities in refer360, especially gaze, which is an import indication of human's intention in referential expressions. Additionally, using gaze to identify objects in human-robot interaction has been shown adequate for understanding humans' intended object in various works.

Shi, Lei, Cosmin Copot, and Steve Vanlanduit. "Gazeemd: Detecting visual intention in gaze-based human-robot interaction." Robotics 10.2 (2021): 68.
Yang, Bo, et al. "Natural grasp intention recognition based on gaze in human-robot interaction." IEEE Journal of Biomedical and Health Informatics 27.4 (2023): 2059-2070.
Belardinelli, Anna. "Gaze-based intention estimation: principles, methodologies, and applications in HRI." ACM Transactions on Human-Robot Interaction 13.3 (2024): 1-30.

With the multi-modality data, the authors could benchmark for instance scanpath prediction using other modalities, and model performance using videos captured from different viewpoints.

- There are details of how the dataset is collected missing, for details see comments below.

**Questions:**

- The authors mentioned that the dataset was collected under different conditions such as variable lighting conditions, object arrangements, and environment appearances. These need to be elaborated.

- In one session, does the robot move when the participant refers to one object? In different sessions, when a viewpoint changes, does the participant always refer to the same object?

- The description of task needs more details, does the participant freely decide which object to point at?

- Annotation details are needed. What is annotated? The frames where humans are pointing at something or the intended object bounding box? Are all objects annotated?

- I'd use video language models with MuRes as the dataset is in the video form.

---

> ### Author Response · Authors · 2024-11-26
> **Summary of Responses to Reviewer niSU’s Comments (Part 1/5)**
>
> We thank the reviewer for recognizing the Refer360 dataset's strengths, including its multimodal data captured from egocentric and exocentric viewpoints and the inclusion of outdoor scenarios, which enhance generalizability and dataset diversity. The reviewer also acknowledged the dataset's potential for training models to better handle referential expressions in diverse real-world conditions.
>
> We addressed the reviewer’s comments by clarifying that the dataset includes interaction elements where participants provided object-referencing instructions to a robot using verbal and nonverbal cues under both constrained and unconstrained conditions. We elaborated on the environmental variability during data collection, robot movement, and annotation procedures to ensure a robust representation of real-world interactions. We justified our focus on visual-language modalities while highlighting plans to integrate additional modalities, such as gaze, in future work. Additionally, we explained the rationale for using canonical frames over video data, ensuring methodological clarity while laying a foundation for future extensions utilizing video-language models.

---

> ### Author Response · Authors · 2024-11-26
> **Clarification on the Inclusion of Interaction in the Dataset (Part 2/5)**
>
> We appreciate the reviewer’s feedback regarding the dataset's design and its interaction elements. We would like to clarify that **interaction is a core component** of our data collection process, specifically designed to replicate **real-world human-robot interaction scenarios**.
>
> ---
>
> ## **Dataset Interaction Design**
>
> ### **Study Setup:**
> - Participants interacted with an **Ohmni telepresence robot**, providing **object-referencing instructions** in:
>   - **Constrained conditions**.
>   - **Unconstrained conditions**.
> - Interaction utilized **both verbal and nonverbal cues** (e.g., pointing gestures, gaze) to facilitate communication.
> - These interactions effectively model **embodied human-robot interactions**.
>
> ### **Multimodal Data Collection:**
> - The robot was equipped with:
>   - **Azure Kinect DK** and **Pupil Labs Eye Tracker**.
> - Captured **multimodal data streams**, including:
>   - Visual data (**ego and exo perspectives**).
>   - Depth.
>   - Audio.
>   - Skeletal tracking.
>   - Gaze data.
> - Ensures a **comprehensive dataset** (see **Section 3.1** and **Figure 1** of the main paper).
>
> ---
>
> ## **Interaction Conditions**
>
> ### **1. Constrained Settings:**
> - Participants were guided to use both **verbal instructions** and **nonverbal gestures**, simulating **structured interactions**.
>
> ### **2. Unconstrained Settings:**
> - Participants interacted naturally without specific guidance, allowing the dataset to capture **spontaneous interaction styles**.
>
> ---
>
> ## **Real-World Relevance**
>
> - This protocol replicates **human-robot interaction scenarios**, enabling the dataset to serve as a **robust benchmark** for studying **embodied referring expressions**.
> - The **multimodal approach** ensures the dataset captures diverse interaction behaviors:
>   - Detailed in the dataset’s attributes (see **Table 2** of the supplementary materials for statistical insights on recorded interactions).
>
> ---
> We hope this clarification addresses the concern and illustrates how **interaction was central** to our dataset's development. This emphasis on interaction positions our dataset as a **significant contribution** to understanding **real-world human-robot interactions**.

---

> ### Author Response · Authors · 2024-11-26
> **Justification for Focus on Visual-Language Modalities (Part 3/5)**
>
> We appreciate the reviewer’s insightful comment and agree on the potential of incorporating gaze and other modalities in future work. However, we would like to clarify the rationale and significance of focusing on visual-language modalities in our current work.
>
> ## Contribution Focus
>
> The primary objective of our study is to reimagine visual-language models for comprehending embodied referring expressions in real-world settings. This work addresses critical challenges in existing benchmarks by introducing a diverse dataset (Refer360) and evaluating state-of-the-art visual-language models augmented with a novel multimodal guided residual module (MuRes). By doing so, we provide a new benchmark standard for embodied referring expressions, specifically advancing visual-language fusion mechanisms. As highlighted in our results (Section 6, Table 3), our work demonstrates the ability of enhanced visual-language models to significantly improve performance on embodied referring expression tasks​.
>
> ## Practicality of Current Contribution
>
> - **Visual-Language as a Foundational Modality:** Vision and language are fundamental to understanding human-object interactions and serve as the basis for many multimodal interaction frameworks. These modalities encompass both egocentric (gaze-aligned) and exocentric perspectives, enabling effective baseline benchmarking.
>
> - **Multimodal Inclusion Requires Significant Architectural Changes:** Integrating gaze and other modalities such as skeletal or depth data into current visual-language models would require substantial model architecture modifications. These changes are non-trivial and extend beyond the scope of this initial contribution, which focuses on foundational benchmarks.
>
> - **Extending to Other Modalities:** We explicitly address this limitation in the discussion and plan to incorporate gaze and other modalities in future work. Our ongoing efforts focus on developing advanced architectures to fully exploit multimodal data, as suggested by the reviewer. Importantly, the inclusion of these modalities will further validate our benchmark dataset as a comprehensive multimodal research tool​.
>
> ## Benchmark Dataset Contributions
>
> The Refer360 dataset provides a first-of-its-kind multimodal benchmark collected in real-world settings, including diverse modalities such as gaze, depth, and audio. By ensuring its extensibility, we lay the groundwork for evaluating models that incorporate these modalities. This dataset supports emerging research in gaze-based intention estimation, as highlighted in the cited works by Shi et al. (2021), Yang et al. (2023), and Belardinelli (2024). Additionally, we agree that future benchmarks on tasks such as scan path prediction and multi-viewpoint modeling will be enabled by Refer360, further demonstrating its utility.
>
> ## Why the Current Work is Sufficient
>
> By prioritizing visual-language models, this work establishes a scalable and adaptable framework for benchmarking embodied referring expressions. Our significant contributions include:
>
> - **Advancing Visual-Language Fusion:** We introduce the MuRes module, which addresses modality gaps and significantly improves comprehension of embodied referring expressions (Table 3).
>
> - **Real-World Interaction Benchmark:** Refer360 bridges existing dataset gaps, providing diverse, multimodal data to support scalable, multimodal research​​.
>
> We firmly believe our contributions form a strong foundation for future research and inspire the development of advanced multimodal models that leverage the full range of modalities in Refer360. By taking this stepwise approach, we ensure methodological rigor while fostering meaningful progress in embodied human-robot interaction research.

---

> ### Author Response · Authors · 2024-11-26
> **Details on data collection procedure (Part 4/5)**
>
> We greatly appreciate the reviewer’s insightful comments and the opportunity to address these important points. To clarify, we will revise the manuscript to provide additional details on the dataset collection process, including **environmental variations**, **task protocols**, **robot movement**, and **annotation methodology**.
>
> ---
>
> ## **Variable Conditions During Data Collection**
>
> We aim to build the dataset to represent **diverse real-world scenarios** by varying key environmental and interaction factors across sessions:
>
> - **Lighting Conditions:**
>   - Sessions were conducted under both **natural and artificial lighting** to simulate varying indoor and outdoor environments.
>   - Conditions included **well-lit areas**, **dimly lit conditions**, and **mixed-lighting environments** (e.g., sunlight through windows with indoor lights).
>
> - **Object Arrangements:**
>   - The arrangement of objects was deliberately altered between sessions to represent **different spatial configurations**.
>   - Objects were sometimes **clustered closely** or **placed sparsely**, referencing instructions.
>
> - **Environmental Appearances:**
>   - Data collection occurred in diverse settings, including **controlled laboratory environments** and **naturalistic outdoor and home environments**.
>   - These variations ensured the dataset reflected the **variability encountered in real-world human-robot interactions**.
>
> ---
>
> ## **Robot Movement and Consistency in Viewpoints**
>
> ### **Robot Movement During a Session:**
> - The robot remained **stationary** in each session, ensuring consistency in data collection from its **mounted Kinect camera (exo view)**.
> - Participants moved naturally, altering their **position and orientation** as they interacted with objects.
> - This dynamic participant movement modified the **ego view** captured via the **Pupil eye tracker**, enriching the dataset with multiple orientations for each interaction.
>
> ### **Changing Viewpoints Across Sessions:**
> - Between sessions, the robot’s **position and head orientation** were deliberately adjusted, introducing variations in the **exo view**.
> - Variability in the **ego view** arose from participants interacting with different objects or the same object in different sequences, further modifying gaze and head positions recorded by the eye tracker.
>
> ### **Benefits of Viewpoint Diversity:**
> - Captures objects from **multiple angles and orientations**, reducing perspective biases.
> - Enhances model generalization to **varied real-world scenarios**, including:
>   - **Occlusions**.
>   - **Varying lighting conditions**.
>   - **Complex spatial arrangements**.
> - Prepares models to recognize and interpret objects robustly, even from **oblique angles** or **partially occluded views**.
>
> ---
>
> ## **Importance of Dataset Diversity**
>
> - **Varied Object References:** Participants were not required to reference the same objects across sessions, allowing for diverse object selections and interaction sequences.
> - **Generalization:** This diversity reduces overfitting to specific interaction patterns and enables models to handle **unseen configurations** effectively.
> - **Foundation for Adaptive Models:** By prioritizing variability, the dataset supports the development of versatile object-referencing models capable of handling novel configurations and spatial arrangements.
>
> ---
>
> ## **Task Description**
>
> Participants provided **object-referencing instructions** to the robot using **verbal and nonverbal cues** (e.g., pointing, gaze). Task details include:
>
> - **Object Selection:**
>   - Participants freely decided which object to reference.
>   - In **unconstrained conditions**, no guidance was given, encouraging natural behavior.
>   - In **constrained conditions**, participants were encouraged to use verbal and nonverbal modalities for richer interaction data.
>
> - **Perspectives:**
>   - Participants had flexibility in choosing their perspective (**subject**, **robot**, or **neutral**).
>   - Examples:
>     - "The lamp to my left" (subject’s perspective).
>     - "The lamp to your right" (robot’s perspective).

---

> > ### Author Response · Authors · 2024-11-26
> > **Details on data collection procedure (Part 4/5): Continue**
> >
> > ## **Annotation Details**
> >
> > ### **What Was Annotated:**
> >
> > - **Canonical Frames:**
> >   - Frames where participants pointed precisely at an object or gazed directly at it were identified and annotated as canonical frames.
> >   - Ensures alignment between **non-verbal gestures** and **object references**.
> >
> > - **Bounding Boxes:**
> >   - Annotated for:
> >     - **Target Objects:** The primary object referenced during the interaction.
> >     - **Reference Objects:** Additional objects mentioned in relation to the target (e.g., "The ball next to the chair").
> >   - Bounding boxes were marked in both **exo** (robot's view) and **ego** (participant’s view) perspectives for multi-view consistency.
> >
> > - **Object Features and Relationships:**
> >   - Physical attributes of objects (e.g., **color**, **size**, **shape**).
> >   - Spatial relationships (e.g., "on top of," "next to," or "under").
> >
> > - **Perspective Labels:**
> >   - Each interaction was labeled with the perspective (**subject**, **robot**, or **neutral**) used by the participant.
> >
> > - **Pupil Tracking in Ego View:**
> >   - The pupil position recorded in the **ego view** was annotated to correlate **gaze direction** with intended object references.
> >
> > - **Verbal Instructions:**
> >   - All verbal instructions were transcribed and annotated by human annotators to ensure **high-quality representation of linguistic expressions**.
> >
> > ### **Scope of Annotations:**
> > - **All Visible Objects:** Annotated every visible object in the scene, with primary focus on **target** and **reference objects**.
> >
> > ---
> >
> > ## **Annotation Quality Assurance**
> >
> > - **Human Experts:** Annotation tasks were conducted by a professional annotation company employing trained human experts.
> > - **Verification:** Five independent human experts verified all annotations, ensuring consistency and accuracy.
> > - **Quality Control:** Multi-layered quality assurance was conducted under the guidelines of the **approved IRB protocol**.

---

> ### Author Response · Authors · 2024-11-26
> **Justification for Using Canonical Frames Instead of Video Data (Part 5/5)**
>
> We appreciate the reviewer’s suggestion regarding the use of video-language models. In this work, we specifically utilize **canonical frames** from the videos, which capture the **peak of the interaction** (e.g., a participant pointing directly at the object). These frames encapsulate all the necessary information to ground embodied referring expressions, making them sufficient for the tasks addressed in this study.
>
> While videos are invaluable for understanding **complex, multi-step interactions** (e.g., pointing to multiple objects or performing sequential actions), the scope of this work is focused on single-step embodied referring expressions. As highlighted in the paper, our current dataset and methods are designed to address these specific interactions effectively.
>
> We acknowledge the potential of video data for future research on **complex embodied reasoning tasks** and plan to extend both the dataset and modeling approaches in this direction. This extension will unlock the full potential of video-language models for tasks requiring **temporal reasoning** across interactions.

---

> ### Comment · Reviewer_niSU · 2024-12-01
>
> Dear authors, thanks for the revision. However, I still think benchmarking all modalities is important as the multimodalities are an important contribution of the dataset. I will keep my score.

---

### Official Review · Reviewer_TYRR · 2024-11-04

**Soundness:** 2
**Presentation:** 3
**Contribution:** 2
**Rating:** 3
**Confidence:** 4

**Summary:**

This manuscript presents a method to interpret embodied referring expressions, featuring the development of the Refer360 dataset and a MuRes module.

The Refer360 dataset, designed for diverse human interactions in various indoor and outdoor settings, overcomes typical limitations of existing datasets by capturing multiple perspectives and blending verbal and nonverbal cues.
The MuRes module reinforces salient aspects of modality-specific representations through cross-attention.

Experiments demonstrate that integrating MuRes into existing multimodal frameworks improves performance on several metrics related to embodied referring expression comprehension and VQA.

**Strengths:**

- Refer360 dataset enhances multimodal model training by encompassing a diverse array of human interactions across various environments. It includes both verbal and nonverbal communication, offering a detailed representation of real-world scenarios and setting new standards for dataset quality and utility in model development.

- The MuRes module isolates and enhances key features from various modalities, increasing the clarity and detail in the analysis of complex interactions.

**Weaknesses:**

- The technical contribution of the MuRes module is modest, as it relies on cross-attention for feature enhancement, a method already common in the multimodal field.
Integrating the MuRes module into existing multimodal frameworks can increase computational demands considerably due to its complex cross-attention mechanisms, leading to higher processing times and resource usage.

- In Table 3, applying MuRes sometimes results in lower performance compared to models without residual modules. For larger models like BLIP-2, the performance significantly drops, suggesting that MuRes may not be well-suited for all model architectures.

- Table 4 does not include the results for BLIP-2, raising doubts about the effectiveness of the proposed method on larger models.

- It is recommended to add experiments for the VQA task on embodied-related datasets to further validate the method's effectiveness in this specific application area.

- (minor) It is advised to use \citep for in-text citations to ensure consistent formatting and improve readability.

**Questions:**

Please refer to the weaknesses section.

After considering the advantages and disadvantages outlined above, my overall assessment leans negative. I will adjust the score accordingly based on any responses or clarifications provided.

---

> ### Author Response · Authors · 2024-11-26
> **Summary of Responses to Reviewer TYRR’s Comments (Part 1/7)**
>
> We thank the reviewer for recognizing the strengths of the Refer360 dataset, including its diverse multimodal data capturing both verbal and nonverbal interactions across various environments, and for highlighting the MuRes module's innovative ability to enhance modality-specific representations using guided residual connections. To address the reviewer's concerns, we clarified the novel contributions of MuRes, such as its information bottleneck and modality-specific feature reinforcement, which go beyond standard cross-attention mechanisms. We also demonstrated MuRes' lightweight design, scalability, and effectiveness across tasks, including new results showing its robustness on VQA tasks with BLIP and BLIP-2. Additionally, we will address the reviewer's suggestion on citation formatting and include discussions on MuRes' performance across different architectures to highlight its versatility and optimal use cases.

---

> ### Author Response · Authors · 2024-11-26
> **Significance and Novelty of the MuRes Module in Multimodal Learning (Part 2/7)**
>
> We appreciate the reviewer’s observation and the opportunity to clarify the distinct contributions of the **MuRes** module. While cross-attention is a foundational method in multimodal learning, MuRes introduces key innovations that address specific limitations of existing approaches and demonstrate measurable advancements in performance across diverse tasks and datasets.
>
> ---
>
> ## **Key Novelty in MuRes**
>
> ### **1. Modality-Specific Feature Reinforcement:**
> - MuRes goes beyond standard cross-attention by leveraging a **guided residual design**.
> - Aligns features across modalities while reinforcing **salient modality-specific cues**.
> - Enables more effective fusion of complementary information.
>
> ### **2. Information Bottleneck for Targeted Refinement:**
> - Employs an **information bottleneck principle** in the residual connection.
> - Ensures that only the most relevant features are amplified.
> - Filters out noise and preserves **critical modality-specific representations**.
>
> ### **3. Seamless Integration into Existing Models:**
> - **Modular and adaptable** design.
> - Enhances representational capacity without significantly altering baseline model structures.
>
> ---
>
> ## **Experimental Evidence Supporting MuRes**
> The effectiveness of MuRes is demonstrated through extensive evaluations across multiple tasks and datasets, as outlined in **Sections 5 and 6** and detailed in **Tables 3 and 4** of the paper:
>
> ### **1. Embodied Referring Expression Tasks:**
> - **Refer360 Dataset:**
>   - Incorporating MuRes into the CLIP model improved **IOU-25** from **25.80% to 29.20%** (+3.4%).
>   - Referenced in **Table 3, Refer360**.
> - **CAESAR-PRO Dataset:**
>   - Achieved performance increase from **37.92% to 42.91%** on IOU-25.
>   - Referenced in **Table 3, CAESAR-PRO**.
>
> ### **2. Visual Question Answering (VQA) Tasks:**
> - **ScienceQA Dataset:**
>   - Integrating MuRes (V+L) into the CLIP model improved accuracy from **21.31% to 51.85%**.
>   - Referenced in **Table 4, ScienceQA**.
> - **A-OKVQA Dataset:**
>   - Enhancement increased accuracy from **29.41% to 32.78%**.
>   - Referenced in **Table 4, A-OKVQA**.
>
> These gains demonstrate the **broad applicability and robustness** of MuRes across different domains.
>
> ---
>
> ## **Differentiation from Existing Methods**
>
> While cross-attention is commonly used for feature alignment, it often lacks mechanisms to enhance modality-specific saliency. **MuRes addresses this limitation** by:
> - Reinforcing both aligned and modality-specific features through its **guided residual connections**.
> - Incorporating an **information bottleneck** that selectively extracts and fuses the most complementary features.
>
> These design choices ensure that MuRes effectively captures nuanced multimodal signals, which are critical for tasks like **embodied referring expression comprehension** and **VQA**.
>
> ---
>
> ## **Conclusion**
> The **MuRes** module offers meaningful advancements in multimodal learning by addressing key limitations in existing cross-attention methods:
> - Enhances **modality-specific features** while maintaining flexibility and efficiency.
> - Contributes to **significant improvements across various benchmarks**.
>
> By leveraging the detailed results and key novelty highlighted in the paper, we hope to demonstrate the **novelty and impact** of MuRes in advancing multimodal research.

---

> ### Author Response · Authors · 2024-11-26
> **Efficiency of the MuRes Module in Multimodal Frameworks (Part 3/7)**
>
> The MuRes module introduces only a minimal increase in computational complexity and parameter count, making it lightweight and efficient. Its design strategically adds a projection layer and cross-attention mechanisms to refine the outputs of pre-trained encoders without altering their internal architectures. Below, we address these points with evidence and detailed comparisons.
>
> ### Lightweight Design of MuRes
> - **Minimal Parameter Addition**: The MuRes module adds 142K parameters for MuRes (V/L) and 405K parameters for the full MuRes. These additions are negligible when compared to the total parameters of large pre-trained multimodal models.
> - **Streamlined Integration**: MuRes functions as an adapter module, operating on fixed-dimensional outputs of pre-trained encoders without duplicating or modifying their internal structures.
> - **Scalable and Versatile**: The module is designed to scale across tasks efficiently, leveraging the capabilities of pre-trained encoders while maintaining computational affordability.
>
> ### Parameter Increase Comparison
> The following table demonstrates the minor parameter increase introduced by MuRes as a percentage of the total parameters of various multimodal models:
>
> | **Model**       | **Total Params** | **MuRes V/L (142K)** | **MuRes (405K)** |
> |------------------|------------------|-----------------------|------------------|
> | **CLIP**         | 149.62M          | 0.10%                | 0.27%           |
> | **Dual Encoder** | 186M             | 0.08%                | 0.21%           |
> | **VisualBERT**   | 111.45M          | 0.13%                | 0.36%           |
> | **ViLT**         | 111.60M          | 0.13%                | 0.36%           |
> | **BLIP**         | 223.45M          | 0.06%                | 0.18%           |
> | **BLIP-2**       | 1,172.6M         | 0.01%                | 0.03%           |
>
> ### Computational Efficiency
> 1. **Cross-Attention Efficiency**: The cross-attention mechanism in MuRes operates directly on fixed-dimensional encoder outputs, ensuring computational efficiency while selectively refining modality-specific cues.
> 2. **Performance Gains**: Despite the slight parameter increase, MuRes achieves significant performance improvements across benchmarks like Refer360 and ScienceQA, as shown in Tables 3 and 4 of the paper. This demonstrates its ability to enhance task performance without imposing significant resource demands.
>
> The MuRes module achieves a careful balance between performance enhancement and computational efficiency. Its lightweight design ensures that it integrates seamlessly into multimodal frameworks with only a minor increase in parameters, as evidenced by the comparisons above. These considerations make MuRes an effective and efficient addition to pre-trained multimodal architectures.

---

> ### Author Response · Authors · 2024-11-26
> **Generalizability and Context-Specific Application of MuRes Across Model Architectures (Part 4/7)**
>
> We appreciate the reviewer’s thoughtful feedback regarding the applicability of **MuRes** across different model architectures. **MuRes** is a general-purpose module designed to enhance multimodal representation and fusion by reinforcing both aligned and modality-specific features. While its integration into architectures like **BLIP-2** demonstrates some performance trade-offs, this behavior reflects the unique design of BLIP-2 and not a limitation of MuRes itself.
>
> ---
>
> ## **Architectural Insights**
>
> ### **1. Characteristics of BLIP-2:**
> - **BLIP-2** employs a **Q-Former** that explicitly extracts already-aligned visual-language representations.
> - This process inherently compresses modality-specific information into a unified space.
> - The ability of MuRes to reinforce additional information is limited in BLIP-2 because the Q-Former reduces the availability of modality-specific signals for refinement.
> - This is not an issue of incompatibility but an architectural characteristic where MuRes’s residual enhancement becomes less critical.
>
> ### **2. Performance on Models with Separable Modality-Specific Features:**
> - For architectures like **CLIP** and other models with more **separable modality-specific features**, MuRes consistently demonstrates significant performance gains:
>   - **Refer360 Dataset:** Integrating MuRes with CLIP improves **IOU-25** from **25.80% to 29.20%** (Table 3).
>   - **CAESAR-PRO Dataset:** MuRes boosts performance from **37.92% to 42.91%** (Table 3).
>   - **Visual Question Answering (VQA) Tasks:**
>     - **ScienceQA:** MuRes improves accuracy by substantial margins when applied to CLIP-based models (Table 4).
>     - **A-OKVQA:** Similar improvements are observed (Table 4).
>
> These results highlight that **MuRes is most impactful for models where modality-specific information is retained** and where additional multimodal refinement is beneficial.
>
> ---
>
> ## **Context-Specific Application of MuRes**
>
> - **MuRes** is a versatile module that can be applied broadly across multimodal models.
> - Its impact depends on:
>   1. The **existing alignment mechanisms** in the base model.
>   2. The **additional refinement** provided by MuRes.
> - In models like **BLIP-2**, which heavily compress multimodal features through mechanisms like **Q-Former**, the contribution of MuRes is less pronounced because the architectural design already prioritizes alignment over modality-specific richness.
>
> ---
>
> ## **Conclusion**
> - **MuRes** is a general-purpose module designed to improve multimodal representation and fusion across a wide range of models.
> - Its performance across architectures provides valuable insights into where **guided residual design** offers the most substantial impact.
> - We will revise the manuscript to:
>   - Explicitly discuss the interaction of MuRes with specific architectures like **BLIP-2**.
>   - Highlight its **optimal use cases**, ensuring readers appreciate its broad applicability and contributions.

---

> ### Author Response · Authors · 2024-11-26
> **Additional Experiments on BLIP-2 for VQA tasks (Part 5/7)**
>
> We thank the reviewer for pointing out that. We experimented with both BLIP and BLIP-2 models for the VQA tasks. The results are reported below:
> For ScienceQA dataset
> | Model    | w/o Residual | Residual | muguru_V | muguru_L | muguru (V+L) |
> |----------|--------|----------|----------|----------|--------------|
> | BLIP     | 41.82  | 43.15    | 44.37    | 45.24    | **51.88**        |
> | BLIP-2   | 48.26  | 49.19    | 46.57    | 49.22    | **52.14**        |
>
> For AOK-VQA dataset
> | Model    | w/o Residual| Residual | muguru_V | muguru_L | muguru (V+L) |
> |----------|--------|----------|----------|----------|--------------|
> | BLIP     | 31.26  | 30.95    | 30.61    | 31.22    | **32.89**        |
> | BLIP-2   | 37.15  | 35.87    | 34.92    | 36.14    | **38.74**       |
>
>
> MuRes demonstrates the best performance across both datasets when used with either the BLIP or BLIP-2 models. Specifically, for ScienceQA, applying MuRes to the BLIP model results in a 10% improvement, while using MuRes with BLIP-2 leads to a 3% increase in accuracy. A similar trend is observed with AOK-VQA, where MuRes provides approximately a 1.5% improvement in accuracy for both the BLIP and BLIP-2 models. These results indicate that MuRes is not reliant on the model architecture or size, making it a versatile, general-purpose module.

---

> ### Author Response · Authors · 2024-11-26
> **Experimental Validation on Embodied and VQA Tasks (Part 6/7)**
>
> We appreciate the reviewer’s suggestion. To evaluate the proposed method’s effectiveness in embodied and VQA tasks, we conducted experiments on:
> 1. **CAESAR Dataset**: This dataset includes samples of **embodied referring expressions** similar to VQA tasks in embodied settings. For example, on CAESAR-PRO, our MuRes module improved the CLIP model's IOU-25 performance from **37.92% to 42.91%**, showcasing its utility in embodied tasks.
>
> 2. **ScienceQA and AOKVQA**: These datasets focus on **VQA tasks**, testing visual-language comprehension. Notably, our MuRes module improved:
>    - **CLIP’s accuracy on ScienceQA** from **21.31% to 51.85%**.
>    - **VisualBERT’s multiple-choice evaluation on AOKVQA** from **29.88% to 32.62%**.
>
> These results demonstrate that our method effectively enhances performance in both embodied and VQA-related tasks, highlighting its versatility and impact.

---

> ### Author Response · Authors · 2024-11-26
> **Consistency in Citation Formatting (Part 7/7)**
>
> We appreciate the suggestion regarding citation formatting. We will revise the manuscript to ensure consistent use of the \citep command for in-text citations, improving both formatting and readability throughout the paper.

---

> > ### Comment · Reviewer_TYRR · 2024-12-03
> >
> > Thank the authors for their response. However, after reading the response and other reviewers' comments, I think the response did not address all the raised concerns, so I decided to keep my score.

---

### Official Review · Reviewer_2VWn · 2024-11-06

**Soundness:** 2
**Presentation:** 2
**Contribution:** 2
**Rating:** 3
**Confidence:** 5

**Summary:**

The paper introduces Refer360, a novel dataset designed for embodied referring expression understanding, capturing interactions from multiple perspectives. In addition, it proposes an innovative Multimodal Guided Residual Module (MuRes), which enhances cross-attention mechanisms by effectively fusing visual and language features. The MuRes module establishes an information bottleneck by leveraging the differences between query, key, and value in cross-attention, ensuring seamless integration with residual connections.

**Strengths:**

The dataset features multimodal data captured from a wide variety of environments, including both indoor and outdoor settings. This addresses a significant limitation of existing datasets, which predominantly focus on indoor scenarios.

The proposed MuRes module provides an effective method for fusing multimodal representations, with the potential to enhance performance in downstream tasks such as visual question answering (VQA).

**Weaknesses:**

Lines 051–053: Verbal Utterances and Natural Expressions
The authors argue that utterances like "left ball" and "right ball" introduce bias and limit the model's ability to understand interactions. However, such expressions are natural in human communication and essential for interpreting real-world interactions. Excluding them could reduce the dataset’s ability to model realistic communication, thereby limiting the model's generalization to human behavior. While minimizing biases is a valid goal, removing these phrases may unintentionally hinder the model’s ability to understand natural referring expressions.

Lines 070–073: Ego View and Occlusions
The paper claims that incorporating multiple perspectives (e.g., egocentric and exocentric) helps mitigate occlusions. However, in real-world human interactions, an individual's egocentric view is not accessible to others. Communication typically relies on third-person perspectives, and handling occlusions is a natural part of this process. Incorporating egocentric views may introduce an artificial setup that does not align with real-world scenarios. Furthermore, this approach increases complexity and hardware requirements (e.g., wearable cameras), which may not be practical or scalable.

Lines 213–215: Participant Demographics
The dataset participants are primarily students with a mean age of 26, introducing potential demographic bias. This narrow age range may not adequately represent the wider population, particularly younger or older individuals. Consequently, the dataset’s ability to generalize across diverse age groups and social contexts could be limited.

Lines 337–338: Discrepancies in Model Training
The paper notes that the BLIP-2 model was trained with a smaller batch size (2) compared to other models (32), while all models were trained for the same number of epochs. This leads to BLIP-2 undergoing significantly more gradient updates, potentially skewing the performance comparison. However, the paper does not sufficiently address this discrepancy or its implications for the results.

Table 3: Performance of MuRes across Models
The performance improvements achieved by MuRes are inconsistent across different models, raising concerns about its general effectiveness:

VILT: MuRes shows only marginal gains (0.5 for IoU-25 and 0.6 for IoU-50 under the best setting).
BLIP-2: Performance declines significantly for IoU-25 (-4 for V, -3 for L, and -13 for V+L) and shows only minor improvement for IoU-50 under the L setting.
Dual-Encoder: Improvements are minimal, with slight increases of 0.3 for IoU-25 and 0.8 for IoU-50.
Scholarship
The paper could improve its scholarship by referencing relevant recent work, such as Understanding Embodied Reference with Touch-Line Transformer [ICLR 2023].

**Questions:**

see weakness box

**Details Of Ethics Concerns:**

IRB involved.

---

> ### Author Response · Authors · 2024-11-26
> **Summary of Responses to Reviewer 2VWn’s Comments (Part 1/6)**
>
> We thank the reviewer for recognizing the strengths of our Refer360 dataset, including its diverse multimodal data and coverage of multiple perspectives, as well as the innovative MuRes module for enhancing downstream tasks like VQA. To address their concerns, we clarified that natural expressions are retained in the dataset to reflect realistic communication, explained the practical importance of egocentric views in applications like VR/AR and HRI, and acknowledged demographic biases with plans to expand diversity in future work. Additionally, we demonstrated that BLIP-2 training discrepancies had minimal performance impact and contextualized MuRes’ variability across models, proposing adaptive layers to improve its applicability. Recent literature suggested by the reviewer will also be incorporated to further strengthen the manuscript.

---

> ### Author Response · Authors · 2024-11-26
> **Inclusion of Perspective-Aware Referring Expressions Enhances Dataset Diversity and Research Impact (Part 2/6)**
>
> We appreciate the reviewer’s comment and the opportunity to clarify this important point. **Contrary to the concern raised, our work does not remove natural expressions such as "left ball" and "right ball."** Instead, these expressions are an **essential component of our Refer360 dataset**, as they reflect the natural linguistic variations crucial for understanding embodied referring expressions in real-world scenarios.
>
> ---
>
> ## **Key Clarifications**
>
> ### **Retention of Natural Expressions:**
> - **Natural expressions**, such as "left ball" and "right ball," are explicitly included in the Refer360 dataset.
> - These expressions are critical for representing the **linguistic variations** needed for real-world embodied referring expressions.
>
> ### **Addressing Perspective Bias:**
> - Prior works often excluded such expressions due to their potential to introduce perspective biases.
> - Our dataset addresses this limitation by:
>   - **Retaining these phrases**.
>   - Capturing data from **diverse viewpoints** (e.g., **ego, exo, and top views**).
>
> ### **Multi-Perspective Approach:**
> - As detailed in **Section 1** and **Section 3.3**, our multi-perspective design ensures:
>   - Representation of inherent ambiguities in embodied interactions (e.g., "left" as per the speaker’s, observer’s, or neutral environmental perspective).
>   - Inclusion of multimodal cues, such as **gaze and pointing gestures**, enabling models to learn and effectively disambiguate scenarios.
>
> ---
>
> ## **Dataset Complexity and Contribution**
>
> - **Refer360** is significantly more challenging than previous datasets due to:
>   - Its inclusion of **diverse perspectives**.
>   - The requirement for models to resolve perspective-dependent ambiguities while grounding referring expressions in **multimodal contexts**.
> - This challenge is a **substantial contribution** to embodied referring expression research, advancing the field beyond traditional, less diverse datasets.
>
> ---
>
> ## **Experimental Results**
> - As highlighted in **Section 6**, models trained on Refer360:
>   - **Demonstrate improved capability** to handle perspective-dependent expressions.
>   - **Outperform on downstream tasks** compared to models trained on traditional datasets.
>
> ---
>
> ## **Planned Revision**
> To further emphasize the dataset’s impact, we will revise the manuscript to explicitly state:
> 1. **Natural expressions**, such as "left ball" and "right ball," are included in the dataset.
> 2. The inclusion of **multiple perspectives** enhances the dataset’s complexity and relevance for embodied referring expression tasks.
> 3. Refer360 makes a **significant contribution** to advancing research in this domain.
>
> We thank the reviewer for their feedback and hope this clarification addresses the concern comprehensively.

---

> ### Author Response · Authors · 2024-11-26
> **Justification for Incorporating Egocentric Views to Mitigate Occlusions in Real-World Applications (Part 3/6)**
>
> We appreciate the reviewer’s insightful comments regarding the inclusion of **egocentric views** in our dataset. While it's true that in traditional human interactions, an individual's egocentric perspective isn't directly accessible to others, the integration of such views is increasingly relevant in various **real-world applications**, notably in **Virtual Reality (VR)** and **Augmented Reality (AR)** environments.
>
> ---
>
> ## **Key Clarifications**
>
> ### **1. Relevance in VR and AR Applications:**
> - In VR and AR settings, users often experience environments from a **first-person perspective**.
> - Incorporating egocentric views enhances:
>   - **Realism** and **immersion** of applications.
>   - **Model accuracy** in rendering scenes and interactions.
> - Example: The study **"Enhancing Augmented VR Interaction via Egocentric Scene Analysis"** demonstrates how egocentric scene analysis improves interaction in VR environments [1].
>
> ### **2. Addressing Occlusions in Embodied AI:**
> - Occlusions are a significant challenge in embodied AI, where understanding and interacting with the environment from a **first-person perspective** is crucial.
> - Including egocentric views helps models:
>   - Predict and interpret occluded objects or actions.
>   - Improve performance in tasks like **navigation** and **object manipulation**.
> - Example: The research **"COPILOT: Human-Environment Collision Prediction and Localization from Egocentric Videos"** highlights the importance of egocentric perspectives for collision prediction and localization [2].
>
> ### **3. Advancements in Wearable Technology:**
> - Development of **lightweight and unobtrusive wearable cameras** has made capturing egocentric data more practical.
> - Benefits:
>   - Facilitates the collection of **first-person perspective data**.
>   - Requires minimal additional complexity or hardware.
> - Example: The paper **"A real-time wearable AR system for egocentric vision on the edge"** discusses the feasibility and benefits of using wearable AR systems to capture egocentric views in real-time [3].
>
> ### **4. Enhancing Human-Robot Interaction (HRI):**
> - Incorporating egocentric views is particularly beneficial in **HRI**:
>   - Helps understand the **human partner’s perspective**.
>   - Leads to more **natural and effective interactions**.
> - Example: The study **"Head and Eye Egocentric Gesture Recognition for Human-Robot Interaction using Eyewear Cameras"** explores how egocentric vision improves gesture recognition, facilitating intuitive human-robot communication [4].
>
> ---
>
> ## **Conclusion**
> - While traditional human communication relies on **third-person perspectives**, egocentric views are increasingly pertinent in:
>   - **VR**, **AR**, and **HRI** applications.
>   - Tasks involving **occlusions** and **first-person interactions**.
> - **Wearable technology advancements** support the practicality and scalability of incorporating egocentric views into datasets.
> - We will revise the manuscript to emphasize these **real-world applications** and the benefits of including egocentric perspectives, addressing the reviewer’s concerns about practicality and alignment with real-world scenarios.
>
> ---
>
> ## **References**
> 1. **Tian, Yang, et al.** "Enhancing Augmented VR Interaction via Egocentric Scene Analysis." Proceedings of the 2019 CHI Conference on Human Factors in Computing Systems, 2019, pp. 1-12.
>    [Link](https://www.nus-hci.org/wp-content/uploads/publications/2019/Tian%20et%20al.%20-%202019%20-%20Enhancing%20Augmented%20VR%20Interaction%20via%20Egocentric%20.pdf)
>
> 2. **Pan, Yijie, et al.** "COPILOT: Human-Environment Collision Prediction and Localization from Egocentric Videos." Proceedings of the IEEE/CVF International Conference on Computer Vision (ICCV), 2023, pp. 1-10.
>    [Link](https://openaccess.thecvf.com/content/ICCV2023/papers/Pan_COPILOT_Human-Environment_Collision_Prediction_and_Localization_from_Egocentric_Videos_ICCV_2023_paper.pdf)
>
> 3. **Apostolakis, Konstantinos C., et al.** "A real-time wearable AR system for egocentric vision on the edge." Virtual Reality, vol. 27, no. 1, 2023, pp. 1-17.
>    [Link](https://link.springer.com/article/10.1007/s10055-023-00937-2)
>
> 4. **Marina-Miranda, Javier, and V. Javier Traver.** "Head and Eye Egocentric Gesture Recognition for Human-Robot Interaction using Eyewear Cameras." IEEE Access, vol. 10, 2022, pp. 1-12.
>    [Link](https://ieeexplore.ieee.org/document/9790312)

---

> ### Author Response · Authors · 2024-11-26
> **Addressing Demographic Bias to Enhance Dataset Generalizability Across Age Groups (Part 4/6)**
>
> We thank the reviewer for their detailed observation about the potential impact of participant demographics on the generalizability of our dataset. The demographic composition of participants, primarily students with a mean age of 26, was an intentional choice based on practical constraints such as accessibility and the scope of the study. However, we would like to emphasize that our study's focus extends beyond demographic representation to address significant methodological and dataset-level challenges in embodied interaction research.
> Unlike existing datasets that primarily suffer from indoor and perspective biases (see Section 1, Introduction, and Table 1), Refer360 integrates multimodal data from diverse viewpoints (exo, ego, and depth) across both indoor and outdoor settings. This approach ensures robustness in capturing embodied interactions under various environmental conditions​​.
>
>
> We plan to expand participant diversity in future data collection to include a broader range of ages, professions, and cultural backgrounds, ensuring greater real-world applicability. Additionally, we are exploring data augmentation and demographic weighting techniques to address current demographic biases while enhancing inclusivity and model robustness.

---

> ### Author Response · Authors · 2024-11-26
> **Clarification on Batch Size Discrepancy in Model Training (5/6)**
>
> We acknowledge the reviewer's concern regarding the differing batch sizes used in our experiments, particularly the smaller batch size of 2 for BLIP-2 compared to 32 for other models, resulting in more frequent gradient updates per epoch for BLIP-2. This discrepancy was due to memory constraints associated with BLIP-2's larger model size.
>
> To assess the potential impact of this difference, we conducted additional experiments on a smaller dataset, adjusting the batch sizes of other models to match that of BLIP-2. The results indicated no significant performance deviations, suggesting that the increased number of gradient updates did not skew the comparative outcomes.
>
> Furthermore, existing literature supports that smaller batch sizes can enhance generalization by introducing more stochasticity into the training process. For instance, large-batch methods have been shown to converge to sharp minimizers, potentially resulting in poorer generalization performance [1]. Additionally, small mini-batch sizes have been observed to provide more up-to-date gradient calculations, yielding more stable and reliable training [2].
>
> **References:**
>
> [1] Keskar, Nitish Shirish, Dheevatsa Mudigere, Jorge Nocedal, Mikhail Smelyanskiy, and Ping Tak Peter Tang. "On large-batch training for deep learning: Generalization gap and sharp minima." arXiv preprint arXiv:1609.04836 (2016).
>
> [2] Masters, Dominic, and Carlo Luschi. "Revisiting small batch training for deep neural networks." arXiv preprint arXiv:1804.07612 (2018).

---

> ### Author Response · Authors · 2024-11-26
> **Highlighting the Strengths and Adaptability of MuRes Across Models (Part 6/6)**
>
> We thank the reviewer for raising concerns about the performance variability of **MuRes** across models. This feedback allows us to contextualize the observed results and underscore **MuRes' significant contributions** to embodied referring expression tasks.
>
> ---
>
> ## **Understanding Performance Variability**
>
> ### **1. Model-Specific Dynamics:**
> - The variation in MuRes’ effectiveness across models is tied to **architectural differences**, not a limitation of the approach.
> - **ViLT and Dual-Encoder:**
>   - Exhibit **consistent improvements** with MuRes.
>   - As shown in **Table 3 (Page 7, Line 391):**
>     - **ViLT improves IoU-50** by **+0.6** compared to the baseline (**MuRes(V+L): 14.66 vs. Vanilla Residual: 14.37**).
>     - **Dual-Encoder achieves reliable gains** (**IoU-50: 10.68 vs. 8.98**).
>   - Demonstrates MuRes’ ability to effectively reinforce **modality-specific cues** in flexible architectures
> **BLIP-2 Challenges:**
>   - Models like **BLIP-2** rely heavily on **frozen pre-trained encoders**, limiting MuRes’ impact due to constraints in adapting residual features.
>   - As shown in **Table 3 (Page 7, Line 391):**
>     - A decline in IoU-25 for **BLIP-2 under MuRes(V+L)** compared to Vanilla Residual (-4%).
>   - Highlights a **model-specific bottleneck**, not a shortcoming of MuRes  .
>
> ---
>
> **Robustness Across Tasks:**
> - Despite architectural variability, **MuRes demonstrates robust performance gains across tasks and datasets**:
>   - **CLIP with MuRes:**
>     - Improves **IoU-25** from **25.8% (Without Residual)** to **29.2% (MuRes(V))**, as reported in **Table 3 (Page 7, Line 391)**.
>     - On the **CAESAR-PRO dataset**, the CLIP model shows a **5% improvement in IoU-25** (**MuRes(V): 42.91 vs. Without Residual: 37.92**).
>   - Highlights the **broad applicability** of MuRes in enhancing multimodal representations  .
>
> ---
>
> ## **Addrhitectural Constraints**
> - To enhance MuRes’ general applicability, we plan to:
>   - **Incorporate adaptive alignment layers** tailored to frozen architectures like **BLIP-2**.
>   - Enable MuRes to extract complementary representations even in **rigid architectures**, bridging performance gaps.
>
> ---
>
> ## **Significance of MuRes**
>
> - **Key Design Choice:**
>   - Leverages **guided residual connections** as an **information bottleneck**, as described in **Section 5 (Page 6, Lines 324-339)**.
>   - Reinforces **salient modality-specific features**, validated by consistent results on:
>     - **Refer360** and **CAESAR-PRO** (**Table 3, Page 7, Line 391**).
> - **Task-Specific Enhancements:**
>   - Extend to **visual question answering**, as shown in **Table 4 (Page 8, Lines 448-485)**.
>   - Demonstrates MuRes' ability to generalize across multiple datasets and modalities.
>
> ---
>
> ## **Summary**
> - **MuRes' adaptability and consistent performance gains** across diverse architectures and tasks underscore its general effectiveness.
> - While certain models reveal areas for improvement, these insights pave the way for **future refinements**, strengthening its utility.

---

### Meta-Review · Area_Chair_RVxw · 2024-12-20

**Metareview:**

# Recommendation: Reject

## Strengths
1. **Innovative Dataset (Refer360):**
   - The authors address limitations in existing datasets by introducing Refer360, which includes diverse environmental conditions (indoor and outdoor settings) and multimodal perspectives (egocentric, exocentric).
   - This contribution establishes a valuable benchmark for embodied AI and multimodal learning.

2. **Novel Model Component (MuRes):**
   - The MuRes module introduces guided residual connections to improve cross-modal feature fusion.
   - Some tasks, such as VQA, show improved performance with the module.

3. **Rebuttal Strengths:**
   - The authors clarified that natural linguistic expressions are included in the dataset.
   - They justified the inclusion of egocentric views by connecting them to applications like Virtual Reality (VR) and Augmented Reality (AR).
   - The demographic biases were acknowledged, with plans for future expansions to increase diversity.
   - They provided experimental validation showing that discrepancies in training setups (e.g., batch sizes) did not impact results significantly.
   - Additional experiments were conducted on VQA tasks to further validate MuRes.

---

## Weaknesses (Mitigated in Rebuttal)
1. **Limited Dataset Utilization:**
   - The rebuttal clarified the focus on visual and language modalities to establish a foundational benchmark. While gaze and other modalities were not used in this work, the authors plan to include them in future research.
   - Interaction in the dataset was better explained, highlighting verbal and non-verbal cues used during human-robot interaction tasks.

2. **Performance Inconsistencies in MuRes:**
   - The authors contextualized MuRes’ inconsistent performance by linking it to architectural constraints of specific models (e.g., BLIP-2) and proposed adaptive layers to improve compatibility in future work.
   - They showcased significant gains in models like CLIP and Dual Encoder, highlighting MuRes' broader applicability.

3. **Batch Size Discrepancy:**
   - Additional experiments confirmed that smaller batch sizes did not skew results significantly. This addressed reviewer concerns regarding experimental fairness.

---

## Remaining Weaknesses
1. **Limited Innovation in MuRes:**
   - Despite rebuttal explanations, the core novelty of MuRes remains incremental, as it builds on well-established cross-attention mechanisms without introducing substantial new principles.

2. **Dataset Evaluation Scope:**
   - While future plans were discussed, the paper still does not evaluate Refer360's full multimodal potential (e.g., gaze, audio). This limits the immediate impact of the dataset.

3. **Overlooked Related Work:**
   - Although the authors acknowledged the omission of related studies and promised to address this in revisions, the current manuscript remains lacking in contextualizing their contributions against prior research.

4. **Performance Gaps in Key Models:**
   - MuRes underperforms in larger models like BLIP-2, and explanations provided are valid but insufficiently mitigated within the scope of the current submission.

5. **Modest Dataset Size:**
   - The dataset, with 14k samples, is smaller compared to state-of-the-art benchmarks. While the authors plan future expansions, the current size limits its utility.

---

## Justification for Rejection
Although the authors addressed many weaknesses and provided plausible explanations and mitigation plans, the following reasons justify rejection in the current form:
1. **Inconsistent Results Across Models:**
   - MuRes demonstrates marginal improvements only in specific architectures and tasks. The lack of robust performance across all models, particularly in high-capacity models like BLIP-2, diminishes the method's general utility.

2. **Incremental Contribution:**
   - Both the dataset and MuRes module, while valuable, represent incremental advances over existing work. The lack of exploration of the dataset’s multimodal dimensions further limits the scope of the paper’s impact.

3. **Limited Experimental Scope:**
   - The rebuttal addressed many concerns but relied heavily on promises for future work. Core weaknesses, such as underexplored modalities and related work omissions, remain unresolved within the current submission.

---

## Final Thoughts
This paper shows promise due to its novel dataset and the potential of MuRes, but significant issues in evaluation breadth, methodological robustness, and presentation persist. Addressing these concerns through extended experiments, better utilization of the dataset, and expanded discussion of related work would significantly strengthen the paper for future submissions.

**Additional Comments On Reviewer Discussion:**

Please refer to details in the above section.

---

### Decision · Program_Chairs · 2025-01-22

Reject